# Multiple Descent: Design Your Own Generalization Curve

**Lin Chen**
Simons Institute for the Theory of Computing
University of California, Berkeley
CA 94720
lin.chen@berkeley.edu

**Yifei Min**
Department of Statistics and Data Science
Yale University
CT 06511
yifei.min@yale.edu

**Mikhail Belkin**
Halıcıoğlu Data Science Institute
University of California, San Diego
CA 92093
mbelkin@ucsd.edu

**Amin Karbasi**
School of Engineering and Applied Science
Yale University
CT 06511
amin.karbasi@yale.edu

## Abstract

This paper explores the generalization loss of linear regression in variably parameterized families of models, both under-parameterized and over-parameterized. We show that the generalization curve can have an arbitrary number of peaks, and moreover, locations of those peaks can be explicitly controlled. Our results highlight the fact that both classical U-shaped generalization curve and the recently observed double descent curve are not intrinsic properties of the model family. Instead, their emergence is due to the interaction between the properties of the data and the inductive biases of learning algorithms.

## 1 Introduction

The main goal of machine learning methods is to provide an accurate out-of-sample prediction, known as generalization. For a fixed family of models, a common way to select a model from this family is through empirical risk minimization, i.e., algorithmically selecting models that minimize the risk on the training dataset. Given a variably parameterized family of models, the statistical learning theory aims to identify the dependence between model complexity and model performance. The empirical risk usually decreases monotonically as the model complexity increases, and achieves its minimum when the model is rich enough to interpolate the training data, resulting in zero (or near-zero) training error. In contrast, the behaviour of the test error as a function of model complexity is far more complicated. Indeed, in this paper we show how to construct a model family for which the generalization curve can be fully controlled (away from the interpolation threshold) in both under-parameterized and over-parameterized regimes. Classical statistical learning theory supports a U-shaped curve of generalization versus model complexity [31, 33]. Under such a framework, the best model is found at the bottom of the U-shaped curve, which corresponds to appropriately balancing under-fitting and over-fitting the training data. From the view of the bias-variance trade-off, a higher model complexity increases the variance while decreasing the bias. A model with an appropriate level of complexity achieves a relatively low bias while still keeping the variance under control. On the other hand, a model that interpolates the training data is deemed to over-fit and tends to worsen the generalization performance due to the soaring variance.

Although classical statistical theory suggests a pattern of behavior for the generalization curve up to the interpolation threshold, it does not describe what happens beyond the interpolation threshold,

35th Conference on Neural Information Processing Systems (NeurIPS 2021).

commonly referred to as the over-parameterized regime. This is the exact regime where many modern machine learning models, especially deep neural networks, achieved remarkable success. Indeed, neural networks generalize well even when the models are so complex that they have the potential to interpolate all the training data points [61, 10, 32, 34].

Modern practitioners commonly deploy deep neural networks with hundreds of millions or even billions of parameters. It has become widely accepted that large models achieve performance superior to small models that may be suggested by the classical U-shaped generalization curve [13, 38, 55, 35, 36]. This indicates that the test error decreases again once model complexity grows beyond the interpolation threshold, resulting in the so called double-descent phenomenon described in [9], which has been broadly supported by empirical evidence [49, 48, 29, 30] and confirmed empirically on modern neural architectures by Nakkiran et al. [46]. On the theoretical side, this phenomenon has been recently addressed by several works on various model settings. In particular, Belkin et al. [11] proved the existence of double-descent phenomenon for linear regression with random feature selection and analyzed the random Fourier feature model [50]. Mei and Montanari [44] also studied the Fourier model and computed the asymptotic test error which captures the double-descent phenomenon. Bartlett et al. [8], Tsigler and Bartlett [56] analyzed and gave explicit conditions for "benign overfitting" in linear and ridge regression, respectively. Caron and Chretien [16] provided a finite sample analysis of the nonlinear function estimation and showed that the parameter learned through empirical risk minimization converges to the true parameter with high probability as the model complexity tends to infinity, implying the existence of double descent. Liu et al. [42] studied the high dimensional kernel ridge regression in the under- and over-parameterized regimes and showed that the risk curve can be double descent, bell-shaped, and monotonically decreasing.

Among all the aforementioned efforts, one particularly interesting question is whether one can observe more than two descents in the generalization curve. d'Ascoli et al. [21] empirically showed a sample-wise triple-descent phenomenon under the random Fourier feature model. Similar triple-descent was also observed for linear regression [47]. More rigorously, Liang et al. [41] presented an upper bound on the risk of the minimum-norm interpolation versus the data dimension in Reproducing Kernel Hilbert Spaces (RKHS), which exhibits multiple descent. However, a multiple-descent upper bound without a properly matching lower bound does not imply the existence of a multiple-descent generalization curve. In this work, we study the multiple descent phenomenon by addressing the following questions:

- Can the existence of a multiple descent generalization curve be rigorously proven?
- Can an arbitrary number of descents occur?
- Can the generalization curve and the locations of descents be designed?

In this paper, we show that the answer to all three of these questions is yes. Further related work is presented in Section 2.

**Our Contribution.** We consider the linear regression model and analyze how the risk changes as the dimension of the data grows. In the linear regression setting, the data dimension is equal to the dimension of the parameter space, which reflects the model complexity. We rigorously show that the multiple descent generalization curve exists under this setting. To our best knowledge, this is the first work proving a multiple descent phenomenon.

Our analysis considers both the underparametrized and overparametrized regimes. In the over-parametrized regime, we show that one can control where a descent or an ascent occurs in the generalization curve. This is realized through our algorithmic construction of a feature-revealing process. To be more specific, we assume that the data is in $\mathbb{R}^D$, where $D$ can be arbitrarily large or even essentially infinite. We view each dimension of the data as a feature. We consider a linear regression problem restricted on the first $d$ features, where $d < D$. New features are revealed by increasing the dimension of the data. We then show that by specifying the distribution of the newly revealed feature to be either a standard Gaussian or a Gaussian mixture, one can determine where an ascent or a descent occurs. In order to create an ascent when a new feature is revealed, it is sufficient that the feature follows a Gaussian mixture distribution. In order to have a descent, it is sufficient that the new feature follows a standard Gaussian distribution. Therefore, in the overparametrized regime, we can fully control the occurrence of a descent and an ascent. As a comparison, in the underparametrized regime, the generalization loss always increases regardless of the feature distribution. Generally speaking, we show that we are able to design the generalization curve.

On the one hand, we show theoretically that the generalization curve is malleable and can be constructed in an arbitrary fashion. On the other hand, we rarely observe complex generalization curves in practice, besides carefully curated constructions. Putting these facts together, we arrive at the conclusion that realistic generalization curves arise from specific interactions between properties of typical data and the inductive biases of algorithms. We should highlight that the nature of these interactions is far from being understood and should be an area of further investigations.

## 2   Related Work

Our work is directly related to the recent line of research in the theoretical understanding of the double descent [11, 34, 60, 44] and the multiple descent phenomenon [41, 39]. Here we briefly discuss some other work that is closely related to this paper.

**Least Square Regression.**   In this paper we focus on the least square linear regression with no regularization. For the regularized least square regression, De Vito et al. [22] proposed a selection procedure for the regularization parameter. Advani and Saxe [1] analyzed the generalization of neural networks with mean squared error under the asymptotic regime where both the sample size and model complexity tend to infinity. Richards et al. [52] proved for least square regression in the asymptotic regime that as the dimension-to-sample-size ratio $d/n$ grows, an additional peak can occur in both the variance and bias due to the covariance structure of the features. As a comparison, in this paper the sample size is fixed and the model complexity increases. Rudi and Rosasco [53] studied kernel ridge regression and gave an upper bound on the number of the random features to reach certain risk level. Our result shows that there exists a natural setting where by manipulating the random features one can control the risk curve.

**Over-Parameterization and Interpolation.**   The double descent occurs when the model complexity reaches and increases beyond the interpolation threshold. Most previous works focused on proving an upper bound or optimal rate for the risk. Caponnetto and De Vito [15] gave the optimal rate for least square ridge regression via careful selection of the regularization parameter. Belkin et al. [12] showed that the optimal rate for risk can be achieved by a model that interpolates the training data. In a series of work on kernel regression with regularization parameter tending to zero (a.k.a. kernel *ridgeless* regression), Rakhlin and Zhai [51] showed that the risk is bounded away from zero when the data dimension is fixed with respect to the sample size. Liang and Rakhlin [40] then considered the case when $d \asymp n$, showed empirically the multiple descent phenomenon and proved a risk upper bound that can be small given favorable data and kernel assumptions. Instead of giving a bound, our paper presents an exact computation of risk in the cases of underparametrized and overparametrized linear regression, and proves the existence of the multiple descent phenomenon. Wyner et al. [59] analyzed AdaBoost and Random Forest from the perspective of interpolation. There has also been a line of work on wide neural networks [4–6, 23, 3, 58, 14, 2, 18, 62, 54].

**Sample-wise Double Descent and Non-monotonicity.**   There has also been recent development beyond the model-complexity double-descent phenomenon. For example, regarding sample-wise non-monotonicity, Nakkiran et al. [46] empirically observed the epoch-wise double-descent and sample-wise non-monotonicity for neural networks. Chen et al. [19] and Min et al. [45] identified and proved the sample-wise double descent under the adversarial training setting, and Javanmard et al. [37] discovered double-descent under adversarially robust linear regression. Loog et al. [43] showed that empirical risk minimization can lead to sample-wise non-monotonicity in the standard linear model setting under various loss functions including the absolute loss and the squared loss, which covers the range from classification to regression. We also refer the reader to their discussion of the earlier work on non-monotonicity of generalization curves. Dar et al. [20] demonstrated the double descent curve of the generalization errors of subspace fitting problems. Fei et al. [28] studied the risk-sample tradeoff in reinforcement learning.

## 3   Preliminaries and Problem Formulation

**Notation.**   For $x \in \mathbb{R}^D$ and $d \leq D$, we let $x[1 : d] \in \mathbb{R}^d$ denote a $d$-dimensional vector with $x[1 : d]_i = x_i$ for all $1 \leq i \leq d$. For a matrix $A \in \mathbb{R}^{n \times d}$, we denote its Moore-Penrose

pseudoinverse by $A^+ \in \mathbb{R}^{d \times n}$ and denote its spectral norm by $\|A\| \triangleq \sup_{x \neq 0} \frac{\|Ax\|_2}{\|x\|_2}$, where $\| \cdot \|_2$ is the Euclidean norm for vectors. If $v$ is a vector, its spectral norm $\|v\|$ agrees with the Euclidean norm $\|v\|_2$. Therefore, we write $\|v\|$ for $\|v\|_2$ to simplify the notation. We use the big O notation $\mathcal{O}$ and write variables in the subscript of $\mathcal{O}$ if the implicit constant depends on them. For example, $\mathcal{O}_{n,d,\sigma}(1)$ is a constant that only depends on $n$, $d$, and $\sigma$. If $f(\sigma)$ and $g(\sigma)$ are functions of $\sigma$, write $f(\sigma) \sim g(\sigma)$ if $\lim \frac{f(\sigma)}{g(\sigma)} = 1$. It will be given in the context how we take the limit.

**Distributions.** Let $\mathcal{N}(\mu, \sigma^2)$ $(\mu, \sigma \in \mathbb{R})$ and $\mathcal{N}(\mu, \Sigma)$ $(\mu \in \mathbb{R}^n, \Sigma \in \mathbb{R}^{n \times n})$ denote the univariate and multivariate Gaussian distributions, respectively, where $\mu \in \mathbb{R}^n$ and $\Sigma \in \mathbb{R}^{n \times n}$ is a positive semi-definite matrix. We define a family of *trimodal* Gaussian mixture distributions as follows

$$\mathcal{N}^{\text{mix}}_{\sigma, \mu} \triangleq \frac{1}{3}\mathcal{N}(0, \sigma^2) + \frac{1}{3}\mathcal{N}(-\mu, \sigma^2) + \frac{1}{3}\mathcal{N}(\mu, \sigma^2).$$

For an illustration, please see Fig. 1.

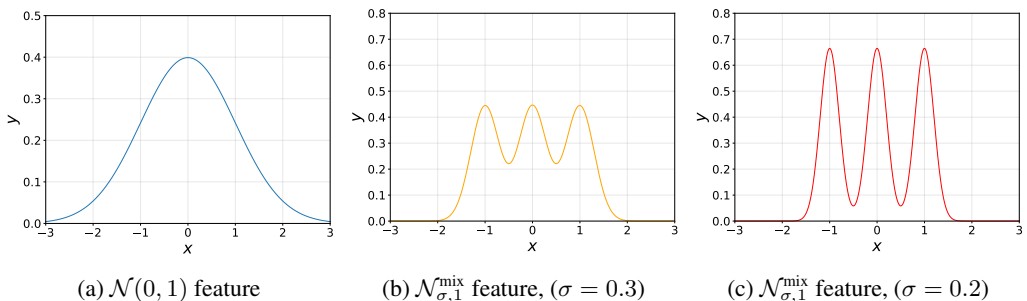

(a) $\mathcal{N}(0, 1)$ feature      (b) $\mathcal{N}^{\text{mix}}_{\sigma, 1}$ feature, $(\sigma = 0.3)$      (c) $\mathcal{N}^{\text{mix}}_{\sigma, 1}$ feature, $(\sigma = 0.2)$

Figure 1: Density functions of the $\mathcal{N}(0, 1)$ and $\mathcal{N}^{\text{mix}}_{\sigma, 1}$ feature. A new entry is independently sampled from the 1-dimensional distribution being either a standard Gaussian or trimodal Gaussian mixture. Smaller $\sigma$ leads to higher concentration around each modes.

Let $\chi^2(k, \lambda)$ denote the noncentral chi-squared distribution with $k$ degrees of freedom and the non-centrality parameter $\lambda$. For example, if $X_i \sim \mathcal{N}(\mu_i, 1)$ (for $i = 1, 2, \ldots, k$) are independent Gaussian random variables, we have $\sum_{i=1}^{k} X_i^2 \sim \chi^2(k, \lambda)$, where $\lambda = \sum_{i=1}^{k} \mu_i^2$. We also denote by $\chi^2(k)$ the (central) chi-squared distribution with $k$ degrees and the $F$-distribution by $F(d_1, d_2)$ where $d_1$ and $d_2$ are the degrees of freedom.

**Problem Setup.** Let $x_1, \ldots, x_n \in \mathbb{R}^D$ be column vectors that represent the training data of size $n$ and let $x_{\text{test}} \in \mathbb{R}^D$ be a column vector that represents the test data. We assume that they are all independently drawn from a distribution

$$x_1, \ldots, x_n, x_{\text{test}} \overset{iid}{\sim} \mathcal{D}.$$

Let us consider a linear regression problem on the first $d$ features, where $d \leq D$ for some arbitrary large $D$. Here, $d$ can be viewed as the number of features revealed. Then the feature vectors are $\tilde{x}_1, \ldots, \tilde{x}_n$, where $\tilde{x}_i = x_i[1 : d] \in \mathbb{R}^d$ denotes the first $d$ entries of $x_i$. The corresponding response variable $y_i$ satisfies

$$y_i = \tilde{x}_i^\top \beta + \varepsilon_i, \quad i = 1, \ldots, n,$$

where the noise $\varepsilon_i \sim \mathcal{N}(0, \eta^2)$. We use the same setup as in [34] (see Equations (1) and (2) in [34]). Moreover, in another closely related work [41], if the kernel is set to the linear kernel, it is equivalent to our setup.

Next, we introduce the estimate $\hat{\beta}$ of $\beta$ and its excess generalization loss. Let $\varepsilon = (\varepsilon_1, \ldots, \varepsilon_n)^\top \in \mathbb{R}^n$ denote the noise vector. The *design* matrix $A$ equals $[\tilde{x}_1, \ldots, \tilde{x}_n]^\top \in \mathbb{R}^{n \times d}$. Let $x = x_{\text{test}}[1 : d]$ denote the first $d$ features of the test data. For the underparametrized regime where $d < n$, the least square solution on the training data is $A^+(A\beta + \varepsilon)$. For the overparametrized regime where $d > n$, $A^+(A\beta + \varepsilon)$ is the minimum-norm solution. In both regimes we consider the solution

$\hat{\beta} \triangleq A^+(A\beta + \varepsilon)$. The excess generalization loss on the test data is then given by

$$
\begin{aligned}
L_d &\triangleq \mathbb{E}\left[\left(y - x^\top \hat{\beta}\right)^2 - \left(y - x^\top \beta\right)^2\right] \\
&= \mathbb{E}\left[\left(x^\top(\hat{\beta} - \beta)\right)^2\right] \\
&= \mathbb{E}\left[\left(x^\top\left((A^+A - I)\beta + A^+\varepsilon\right)\right)^2\right] \\
&= \mathbb{E}\left[(x^\top(A^+A - I)\beta)^2\right] + \mathbb{E}\left[(x^\top A^+\varepsilon)^2\right] \\
&= \mathbb{E}\left[(x^\top(A^+A - I)\beta)^2\right] + \eta^2\mathbb{E}\left\|(A^\top)^+x\right\|^2,
\end{aligned}
\tag{1}
$$

where $y = x^\top\beta + \varepsilon_{\text{test}}$ and $\varepsilon_{\text{test}} \sim \mathcal{N}(0, \eta^2)$. We call the term $\mathbb{E}\left[(x^\top(A^+A - I)\beta)^2\right]$ the *bias* and call the term $\eta^2\mathbb{E}\left\|(A^\top)^+x\right\|^2$ the *variance*.

The next remark shows that in the underparametrized regime, the bias vanishes. The vanishing bias in the underparametrized regime is also observed by Hastie et al. [34] and shown in their Proposition 2.

**Remark 1.** In the underparametrized regime, if $\mathcal{D}$ is a continous distribution (our construction presented later satisfies this condition), the matrix $A$ has independent column almost surely. In this case, we have $A^+A = I$ and therefore the bias $\mathbb{E}\left[(x^\top(A^+A - I)\beta)^2\right]$ vanishes irrespective of $\beta$. In other words, in the underparametrized regime, $L_d$ equals $\eta^2\mathbb{E}\|(A^\top)^+x\|^2$.

According to Remark 1, we have $L_d = \eta^2\mathbb{E}\|(A^\top)^+x\|^2$ in the underparametrized regime. It also holds in the overparametrized regime when $\beta = 0$. Without loss of generality, we assume $\eta = 1$ in the underparametrized regime (for all $\beta$). In the overparametrized regime, we also assume $\eta = 1$ for the $\beta = 0$ case. In this case, we have

$$
L_d = \mathbb{E}\|(A^\top)^+x\|^2.
\tag{2}
$$

We assume a general $\eta$ (i.e., not necessarily being 1) in the overparametrized regime when $\beta$ is non-zero.

We would like to study the change in the loss caused by the growth in the number of features revealed. Recall $L_d = \mathbb{E}\|(A^\top)^+x\|^2$. Once we reveal a new feature, which adds a new row $b^\top$ to $A^\top$ and a new component $a_1$ to $x$, we have $L_{d+1} = \mathbb{E}\left\|\begin{bmatrix}A^\top \\ b^\top\end{bmatrix}^+ \begin{bmatrix}x \\ a_1\end{bmatrix}\right\|^2$.

**Local Maximum and Multiple Descent.** Throughout the paper, we say that a local maximum occurs at a dimension $d \geq 1$ if $L_{d-1} < L_d$ and $L_d > L_{d+1}$. Intuitively, a local maximum occurs if there is an increasing stage of the generalization loss, followed by a decreasing stage, as the dimension $d$ grows. Additionally, we define $L_0 \triangleq -\infty$. If the generalization loss exhibits a single descent, based on our definition, a unique local maximum occurs at $d = 1$. For a double-descent generalization curve, a local maximum occurs at two different dimensions. In general, if we observe local maxima at multiple dimensions, we say there is a multiple descent.

## 4 Underparametrized Regime

First, we present our main theorem for the underparametrized regime below, whose proof is deferred to the end of Section 4. It states that the generalization loss $L_d$ is always non-decreasing as $d$ grows. Moreover, it is possible to have an arbitrarily large ascent, i.e., $L_{d+1} - L_d > C$ for any $C > 0$.

**Theorem 1** (Proof in Section 4.1). *If $d < n$, we have $L_{d+1} \geq L_d$ irrespective of the data distribution. Moreover, for any $C > 0$, there exists a distribution $\mathcal{D}$ such that $L_{d+1} - L_d > C$.*

**Remark 2** ($\mathcal{D}$ can be a product distribution). The first part of Theorem 1 holds irrespective of the data distribution. For the second part of the theorem ( i.e., for any $C > 0$ there exists a distribution such that $L_{d+1} - L_d > C$) to hold, one extremely simple and elegant choice of the distribution $\mathcal{D}$ is a product distribution $\mathcal{D} = \mathcal{D}_1 \times \cdots \times \mathcal{D}_D$ such that $x_{i,j} \stackrel{iid}{\sim} \mathcal{D}_j$ for all $1 \leq i \leq n$, where $\mathcal{D}_j$ is a Gaussian mixture $\mathcal{N}_{\sigma_j,1}^{\text{mix}}$ for some $\sigma_j > 0$. Since the second part of Theorem 1 is of independent interest, the result is summarized by Theorem 4.

**Remark 3** (Kernel regression on Gaussian data). In light of Remark 2, $\mathcal{D}$ can be chosen to be a product distribution that consists $\mathcal{N}_{\sigma_j}^{\mathrm{mix}}$. Note that one can simulate $\mathcal{N}_{\sigma,1}^{\mathrm{mix}}$ with $\mathcal{N}(0,1)$ through the inverse transform sampling. To see this, let $F_{\mathcal{N}(0,1)}$ and $F_{\mathcal{N}_{\sigma,1}^{\mathrm{mix}}}$ be the cdf of $\mathcal{N}(0,1)$ and $\mathcal{N}_{\sigma,1}^{\mathrm{mix}}$, respectively. If $X \sim \mathcal{N}(0,1)$, we have $F_{\mathcal{N}(0,1)}(X) \sim \mathrm{Unif}((0,1))$ and therefore $\varphi_\sigma(X) \triangleq F_{\mathcal{N}_{\sigma,1}^{\mathrm{mix}}}^{-1}(F_{\mathcal{N}(0,1)}(X)) \sim \mathcal{N}_{\sigma,1}^{\mathrm{mix}}$. In fact, we can use a multivariate Gaussian $\mathcal{D}' = \mathcal{N}(0, I_{D \times D})$ and a sequence of non-linear kernels $k^{[1:d]}(x,x') \triangleq \langle \phi^{[1:d]}(x), \phi^{[1:d]}(x') \rangle$, where the feature map is $\phi^{[1:d]}(x) \triangleq [\phi_1(x_1), \phi_2(x_2), \dots, \phi_d(x_d)]^\top \in \mathbb{R}^d$. Here is a simple rule for defining $\phi_j$: if $\mathcal{D}_j = \mathcal{N}_{\sigma_j}^{\mathrm{mix}}$, we set $\phi_j$ to $\varphi_{\sigma_j}$. Thus, the problem becomes a kernel regression problem on the standard Gaussian data.

The first part of Theorem 1, which says that $L_d$ is increasing (or more precisely, non-decreasing), agrees with Figure 1 of [11] and Proposition 2 of [34]. In [34], they proved that the risk increases with $\gamma = d/n$. Note that, at first glance, Theorem 1 may look counterintuitive since it does not obey the classical U-shaped generalization curve. However, we would like to emphasize that the U-shaped curve does not always occur. In Figure 1 and Proposition 2 of these two papers respectively, there is no U-shaped curve. The intuition behind Theorem 1 is that in the underparametrized setting, the bias is always zero and as $d$ approaches $n$, the variance keeps increasing.

Coming to the second part of Theorem 1, we now discuss how we will construct such a distribution $\mathcal{D}$ inductively to satisfy $L_{d+1} - L_d > C$. We fix $d$. Again, denote the first $d$ features of $x_{\mathrm{test}}$ by $x \triangleq x_{\mathrm{test}}[1:d]$. Let us add an additional component to the training data $x_1[1:d], \dots, x_n[1:d]$ and test data $x$ so that the dimension $d$ is incremented by 1. Let $b_i \in \mathbb{R}$ denote the additional component that we add to the vector $x_i$ (so that the new vector is given as $[x_i[1:d]^\top, b_i]^\top$. Similarly, let $a_1 \in \mathbb{R}$ denote the additional component that we add to the test vector $x$. We form the column vector $b = [b_1, \dots, b_n]^\top \in \mathbb{R}^n$ that collects all additional components that we add to the training data.

We consider the change in the generalization loss as follows

$$L_{d+1} - L_d = \mathbb{E}\left[ \left\| \begin{bmatrix} A^\top \\ b^\top \end{bmatrix}^+ \begin{bmatrix} x \\ a_1 \end{bmatrix} \right\|^2 - \left\| (A^+)^\top x \right\|^2 \right]. \tag{3}$$

Note that the components $b_1, \dots, b_n, a_1$ are i.i.d. The proof of Theorem 1 starts with Lemma 2 which relates the pseudo-inverse of $[A,b]^\top$ to that of $A^\top$. In this way, we can decompose $\left\| \begin{bmatrix} A^\top \\ b^\top \end{bmatrix}^+ \begin{bmatrix} x \\ a_1 \end{bmatrix} \right\|^2$ into multiple terms for further careful analysis in the proofs hereinafter.

**Lemma 2** (Proof in Appendix B.1). *Let $A \in \mathbb{R}^{n \times d}$ and $0 \neq b \in \mathbb{R}^{n \times 1}$, where $n \geq d+1$. Additionally, let $P = AA^+$ and $Q = bb^+ = \frac{bb^\top}{\|b\|^2}$, and define $z \triangleq \frac{b^\top(I-P)b}{\|b\|^2}$. If $z \neq 0$ and the columnwise partitioned matrix $[A,b]$ has linearly independent columns, we have*

$$\begin{bmatrix} A^\top \\ b^\top \end{bmatrix}^+ = \left[ \left(I - \frac{bb^\top}{\|b\|^2}\right)\left(I + \frac{AA^+bb^\top}{\|b\|^2 - b^\top AA^+ b}\right)(A^+)^\top, \frac{(I-AA^+)b}{\|b\|^2 - b^\top AA^+ b} \right]$$

$$= \left[ (I-Q)(I + \frac{PQ}{1-\mathrm{tr}(PQ)})(A^+)^\top, \frac{(I-P)b}{b^\top(I-P)b} \right]$$

$$= \left[ (I-Q)(I + \frac{PQ}{z})(A^+)^\top, \frac{(I-P)b}{b^\top(I-P)b} \right].$$

In our construction of $\mathcal{D}$, the components $\mathcal{D}_j$ are all continuous distributions. The matrix $I - P$ is an orthogonal projection matrix and therefore $\mathrm{rank}(I-P) = n - d$. As a result, it holds almost surely that $b \neq 0$, $z \neq 0$, and $[A,b]$ has linearly independent columns. Thus the assumptions of Lemma 2 are satisfied almost surely. In the sequel, we assume that these assumptions are always fulfilled.

Theorem 3 guarantees that if $L_d = \mathbb{E}\left\| (A^+)^\top x \right\|^2$ is finite and the $(d+1)$-th features $b_1, \dots, b_n, a_1$ are i.i.d. sampled from $\mathcal{N}(0,1)$ or $\mathcal{N}_{\sigma,1}^{\mathrm{mix}}$, $L_{d+1} = \mathbb{E}\left\| \begin{bmatrix} A^\top \\ b^\top \end{bmatrix}^+ \begin{bmatrix} x \\ a_1 \end{bmatrix} \right\|^2$ is also finite.

**Theorem 3** (Proof in Appendix B.2). *Let $z$ be as defined in Lemma 2. If $b_1, \ldots, b_n, a_1$ are i.i.d. and follow a distribution with mean zero, conditioned on $A$ and $x$, we have*

$$\mathbb{E}_{b,a_1} \left[ \left\| \begin{bmatrix} A^\top \\ b^\top \end{bmatrix}^+ \begin{bmatrix} x \\ a_1 \end{bmatrix} \right\|^2 \right] \leq \mathbb{E}_{b,a_1} \left[ \frac{1}{z} \left\| (A^+)^\top x \right\|^2 + \frac{a_1^2}{b^\top (I - P) b} \right].$$

*In particular, if $d + 2 < n$ and $b_1, \ldots, b_n, a_1 \overset{iid}{\sim} \mathcal{N}(0,1)$, conditioned on $A$ and $x$, we have*

$$\mathbb{E}_{b,a_1} \left[ \left\| \begin{bmatrix} A^\top \\ b^\top \end{bmatrix}^+ \begin{bmatrix} x \\ a_1 \end{bmatrix} \right\|^2 \right] \leq \frac{(n-2) \left\| (A^+)^\top x \right\|^2 + 1}{n - d - 2}.$$

*If $d + 2 < n$ and $b_1, \ldots, b_n, a_1 \overset{iid}{\sim} \mathcal{N}_{\sigma,1}^{\mathrm{mix}}$, conditioned on $A$ and $x$, we have*

$$\mathbb{E}_{b,a_1} \left\| \begin{bmatrix} A^\top \\ b^\top \end{bmatrix}^+ \begin{bmatrix} x \\ a_1 \end{bmatrix} \right\|^2 \leq \frac{(n - 2 + \sqrt{d}) \left\| (A^+)^\top x \right\|^2 + 2/(3\sigma^2) + 1}{n - d - 2}.$$

Using Theorem 3, we can show inductively (on $d$) that $L_d$ is finite for every $d$. Provided that we are able to guarantee finite $L_1$, Theorem 3 implies that $L_d$ is finite for every $d$ if the components are always sampled from $\mathcal{N}(0,1)$ or $\mathcal{N}_{\sigma,1}^{\mathrm{mix}}$.

Making a large $L_d$ can be achieved by adding an entry sampled from $\mathcal{N}_{\sigma,1}^{\mathrm{mix}}$ when the data dimension increases from $d - 1$ to $d$ in the previous step. Theorem 4 shows that adding a $\mathcal{N}_{\sigma,1}^{\mathrm{mix}}$ feature can increase the loss by arbitrary amount, which in turn implies the second part of Theorem 1.

**Theorem 4** (Proof in Appendix B.4). *For any $C > 0$ and $\mathbb{E} \left\| (A^+)^\top x \right\|^2 < +\infty$, there exists a $\sigma > 0$ such that if $b_1, \ldots, b_n, a_1 \overset{iid}{\sim} \mathcal{N}_{\sigma,1}^{\mathrm{mix}}$, we have*

$$\mathbb{E} \left[ \left\| \begin{bmatrix} A^\top \\ b^\top \end{bmatrix}^+ \begin{bmatrix} x \\ a_1 \end{bmatrix} \right\|^2 - \left\| (A^+)^\top x \right\|^2 \right] > C.$$

We are now ready to prove Theorem 1.

### 4.1 Proof of Theorem 1

*Proof.* We follow the notation convention in (3):

$$L_{d+1} - L_d = \mathbb{E} \left[ \left\| \begin{bmatrix} A^\top \\ b^\top \end{bmatrix}^+ \begin{bmatrix} x \\ a_1 \end{bmatrix} \right\|^2 - \left\| (A^\top)^+ x \right\|^2 \right].$$

Recall $d < n$ and the matrix $B' \triangleq \begin{bmatrix} A^\top \\ b^\top \end{bmatrix}$ is of size $(d + 1) \times n$. Both matrices $B'$ and $B \triangleq A^\top$ are fat matrices. As a result, if $x' \triangleq \begin{bmatrix} x \\ a_1 \end{bmatrix}$, we have

$$\|B'^+ x'\|^2 = \min_{z : B'z = x'} \|z\|^2, \quad \|B^+ x\|^2 = \min_{z : Bz = x} \|z\|^2.$$

Since $\{z \mid B'z = x'\} \subseteq \{z \mid Bz = x\}$, we get $\|B'^+ x'\|^2 \geq \|B^+ x\|^2$. Therefore, we obtain $L_{d+1} \geq L_d$. The second part follows from Theorem 4. $\qquad\square$

**Remark 4.** Remark 2 and the proof of Theorem 4 indicate that $\mathcal{D} = \mathcal{D}_1 \times \cdots \times \mathcal{D}_D$ is a product distribution. The construction in the proof also shows that the generalization curve is determined by the specific choice of the $\mathcal{D}_i$'s. Note that permuting the order of $\mathcal{D}_i$'s is equivalent to changing the order by which the features are being revealed (i.e., permuting the entries of the data $x_i$'s). Therefore, given the same data points $x_1, \cdots, x_n \in \mathbb{R}^D$, one can create different generalization curves simply by changing the order of the feature-revealing process.

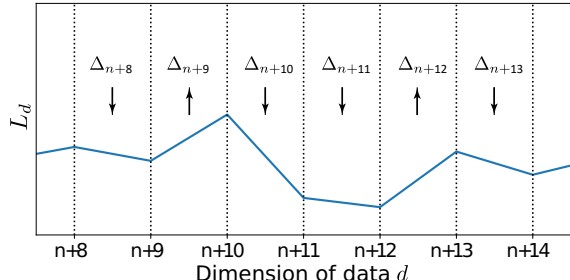

Figure 2: Illustration of the multiple descent phenomenon for the generalization loss $L_d$ versus the dimension of data $d$ in the *overparametrized* regime starting from $d = n+8$. One can fully control the generalization curve to increase or decrease as specified by the sequence $\Delta = \{\downarrow, \uparrow, \downarrow, \downarrow, \uparrow, \downarrow, \dots\}$. Adding a new feature with Gaussian mixture distribution increases the loss, while adding one with Gaussian distribution decreases the loss.

## 5 Overparametrized Regime

In this section, we study the multiple decent phenomenon in the overparametrized regime. Note that as stated in Section 3, we consider the minimum-norm solution here. We first consider the case where the model $\beta = 0$ and $L_d$ is as defined in (2). Then we discuss the setting $\beta \neq 0$.

As stated in the following theorem, we require $d \geq n + 8$. This is merely a technical requirement and we can still say that $d$ starts at roughly the same order as $n$. In other words, the result covers almost the entire spectrum of the overparametrized regime.

**Theorem 5** (Overparametrized regime, $\beta = 0$). *Let $n < D - 9$. Given any sequence $\Delta_{n+8}, \Delta_{n+9}, \dots, \Delta_{D-1}$ where $\Delta_d \in \{\uparrow, \downarrow\}$, there exists a distribution $\mathcal{D}$ such that for every $n + 8 \leq d \leq D - 1$, we have*

$$L_{d+1} \begin{cases} > L_d, & \text{if } \Delta_d = \uparrow \\ < L_d, & \text{if } \Delta_d = \downarrow. \end{cases}$$

In Theorem 5, the sequence $\Delta_{n+8}, \Delta_{n+9}, \cdots, \Delta_{D-1}$ is just used to specify the increasing/decreasing behavior of the $L_d$ sequence for $d > n + 8$. Compared to Theorem 1 for the underparametrized regime, where $L_d$ always increases, Theorem 5 indicates that one is able to fully control both ascents and descents in the overparametrized regime. Fig. 2 is an illustration.

We now present tools for proving Theorem 5. Lemma 6 gives the pseudo-inverse of $A$ when $d > n$.

**Lemma 6** (Proof in Appendix C.1). *Let $A \in \mathbb{R}^{n \times d}$ and $b \in \mathbb{R}^{n \times 1}$, where $n \leq d$. Assume that matrix $A$ and the columnwise partitioned matrix $B \triangleq [A, b]$ have linearly independent rows. Let $G \triangleq (AA^\top)^{-1} \in \mathbb{R}^{n \times n}$ and $u \triangleq \frac{b^\top G}{1 + b^\top G b} \in \mathbb{R}^{1 \times n}$. We have*

$$\begin{bmatrix} A^\top \\ b^\top \end{bmatrix}^+ = \left[ (I - bu)^\top (A^+)^\top, u^\top \right].$$

Lemma 7 establishes finite expectation for several random variables. These finite expectation results are necessary for Theorem 8 and Theorem 9 to hold. Technically, they are the dominating random variables needed in Lebesgue's dominated convergence theorem. Lemma 7 indicates that to guarantee these finite expectations, it suffices to set the first $n + 8$ distributions to the standard normal distribution and then set $\mathcal{D}_{n+8}, \dots, \mathcal{D}_D$ to either a Gaussian or a Gaussian mixture distribution. In fact, in Theorem 8 and Theorem 9, we always add a Gaussian distribution or a Gaussian mixture.

**Lemma 7** (Proof in Appendix C.2). *Let $\mathcal{D} = \mathcal{D}_1 \times \cdots \times \mathcal{D}_D$ be a product distribution where*

(a) $\mathcal{D}_d = \mathcal{N}(0, 1)$ *if $d = 1, \dots, n + 8$; and*

(b) $\mathcal{D}_d$ *is either $\mathcal{N}(0, \sigma_d^2)$ or $\mathcal{N}_{\sigma_d, \mu_d}^{\text{mix}}$ for $d > n + 8$.*

Let $\mathcal{D}_{[1:d]}$ denote $\mathcal{D}_1 \times \cdots \times \mathcal{D}_d$. Assume that every row of $A \in \mathbb{R}^{n \times d}$ and $x \in \mathbb{R}^{d \times 1}$ are i.i.d. and follow $\mathcal{D}_{[1:d]}$. For any $d$ such that $n + 8 \le d \le D$, all of the followings hold:

$$
\begin{aligned}
\mathbb{E}[\|(A^+)^\top x\|^2] &< +\infty, & \mathbb{E}[\lambda_{\max}^2((AA^\top)^{-1})] &< +\infty, \\
\mathbb{E}[\lambda_{\max}((AA^\top)^{-1})\|(A^+)^\top x\|^2] &< +\infty, & \mathbb{E}[\lambda_{\max}^2((AA^\top)^{-1})\|(A^+)^\top x\|^2] &< +\infty.
\end{aligned} \tag{4}
$$

Theorems 8 and 9 are the key technical results for constructing multiple descent in the over-parametrized regime. One can create a descent ($L_{d+1} < L_d$) by adding a Gaussian feature (Theorem 8) and create an ascent ($L_{d+1} > L_d$) by adding a Gaussian mixture feature (Theorem 9).

**Theorem 8** (Proof in Appendix C.3). *If* $\mathbb{E}[\|(A^\top A)^+ x\|^2] > 0$ *and all equations in* (4) *hold, there exists* $\sigma > 0$ *such that if* $a_1, b_1, \ldots, b_n \overset{iid}{\sim} \mathcal{N}(0, \sigma^2)$, *we have*

$$
L_{d+1} - L_d = \mathbb{E}\left\|\begin{bmatrix} A^\top \\ b^\top \end{bmatrix}^+ \begin{bmatrix} x \\ a_1 \end{bmatrix}\right\|^2 - \mathbb{E}\left\|(A^+)^\top x\right\|^2 < 0.
$$

Theorem 9 shows that adding a Gaussian mixture feature can make $L_{d+1} > L_d$.

**Theorem 9** (Proof in Appendix C.4). *Assume* $\mathbb{E}\|(A^+)^\top x\|^2 < +\infty$. *For any* $C > 0$, *there exist* $\mu$, $\sigma > 0$ *such that if* $a_1, b_1, \ldots, b_n \overset{iid}{\sim} \mathcal{N}_{\sigma,\mu}^{\mathrm{mix}}$, *we have*

$$
L_{d+1} - L_d = \mathbb{E}\left\|\begin{bmatrix} A^\top \\ b^\top \end{bmatrix}^+ \begin{bmatrix} x \\ a_1 \end{bmatrix}\right\|^2 - \mathbb{E}\left\|(A^+)^\top x\right\|^2 > C.
$$

The proof of Theorem 5 immediately follows from Theorem 8 and Theorem 9.

*Proof of Theorem 5.* We construct the product distribution $\mathcal{D} = \prod_{d=1}^{D} \mathcal{D}_d$. We set $\mathcal{D}_d = \mathcal{N}(0, 1)$ for $d = 1, \ldots, n + 8$. For $n + 8 < d \le D$, $\mathcal{D}_d$ is either $\mathcal{N}(0, \sigma_d^2)$ or $\mathcal{N}_{\sigma_d, \mu_d}^{\mathrm{mix}}$ depending on $\Delta_d$ being either $\downarrow$ or $\uparrow$.

First we show that for each step $d$, the assumption $\mathbb{E}[\|(A^\top A)^+ x\|^2] > 0$ of Theorem 8 is satisfied. If $\mathbb{E}[\|(A^\top A)^+ x\|^2] = 0$, we know that $(A^\top A)^+ x = 0$ almost surely. Since $\mathcal{D}$ is a continuous distribution, the matrix $A$ has full row rank almost surely. Therefore, $\mathrm{rank}((A^\top A)^+) = \mathrm{rank}(A^\top A) = n$ almost surely. Thus $\dim \ker(A^\top A)^+ = d - n \le d - 1$ almost surely, which implies $x \notin \ker(A^\top A)^+$. In other words, $(A^\top A)^+ x \neq 0$ almost surely. We reach a contradiction. Moreover, by Lemma 7, the assumption $\mathbb{E}\|(A^+)^\top x\|^2 < +\infty$ of Theorem 9 is also satisfied.

If $\Delta_{d-1} = \downarrow$, by Theorem 8, there exists $\sigma_d > 0$ such that if $\mathcal{D}_d = \mathcal{N}(0, \sigma_d^2)$, then $L_d < L_{d-1}$. Similarly if $\Delta_{d-1} = \uparrow$, by Theorem 9, there exists $\sigma_d$ and $\mu_d$ such that $\mathcal{D}_d = \mathcal{N}_{\sigma_d, \mu_d}^{\mathrm{mix}}$ guarantees $L_d > L_{d-1}$.

$\square$

**Gaussian $\beta$ setting.** In what follows, we study the case where the model $\beta$ is non-zero. In particular, we consider a setting where each entry of $\beta$ is i.i.d. $\mathcal{N}(0, \rho^2)$. Recalling (1), define the biases

$$
\mathcal{E}_d \triangleq (x^\top (A^+ A - I)\beta)^2, \quad \mathcal{E}_{d+1} \triangleq \left([x^\top, a_1]([A, b]^+ [A, b] - I) \begin{bmatrix} \beta \\ \beta_1 \end{bmatrix}\right)^2,
$$

and the expected risks

$$
L_d^{\mathrm{exp}} \triangleq \mathbb{E}[\mathcal{E}_d] + \eta^2 \mathbb{E}\left\|(A^\top)^+ x\right\|^2, \quad L_{d+1}^{\mathrm{exp}} \triangleq \mathbb{E}[\mathcal{E}_{d+1}] + \eta^2 \mathbb{E}\left\|\begin{bmatrix} A^\top \\ b^\top \end{bmatrix}^+ \begin{bmatrix} x \\ a_1 \end{bmatrix}\right\|^2, \tag{5}
$$

where $\beta \sim \mathcal{N}(0, \rho^2 I_d)$ and $\beta_1 \sim \mathcal{N}(0, \rho^2)$. The second term in $L_d^{\mathrm{exp}}$ and $L_{d+1}^{\mathrm{exp}}$ is the variance term. Note that $L_d^{\mathrm{exp}}$ is the expected value of $L_d$ in (1) and averages over $\beta$. Theorem 10 shows that one can add a Gaussian mixture feature in order to make $L_{d+1}^{\mathrm{exp}} > L_d^{\mathrm{exp}}$, and add a Gaussian feature in order to make $L_{d+1}^{\mathrm{exp}} < L_d^{\mathrm{exp}}$.

**Theorem 10** (Proof in Appendix C.5). *Let $a_1, \beta_1 \in \mathbb{R}$, $x \in \mathbb{R}^{d \times 1}$, $\beta \in \mathbb{R}^{d \times 1}$, $A \in \mathbb{R}^{n \times d}$ and $b \in \mathbb{R}^{n \times 1}$, where $n \leq d$. Assume that $x, a_1, \beta_1, \beta, A, b$ are jointly independent, $[\beta^\top, \beta_1]^\top \sim \mathcal{N}(0, \rho^2 I_{d+1})$. Moreover, assume that the matrix $[A, b]$ has linearly independent rows almost surely. The following statements hold:*

*(a) If $a_1, b_1, \ldots, b_n \overset{iid}{\sim} \mathcal{N}^{\text{mix}}_{\sigma,\mu}$, for any $C > 0$, there exist $\mu, \sigma$ such that $L^{\text{exp}}_{d+1} - L^{\text{exp}}_d > C$.*

*(b) If $a_1, b_1, \ldots, b_n \overset{iid}{\sim} \mathcal{N}(0, \sigma^2)$, there exists $\sigma > 0$ such that for all*

$$\rho \leq \eta \sqrt{\frac{\mathbb{E}[\|(A^\top A)^+ x\|^2]}{\mathbb{E}\|A^{+\top} x\|^2 + 1}},$$

*we have $L^{\text{exp}}_{d+1} < L^{\text{exp}}_d$.*

Theorem 10 indicates that for $\beta$ obeying a normal distribution, one can still construct a generalization curve as desired by adding a Gaussian or Gaussian mixture feature properly. We make this construction explicit for any desired generalization curve in (the proof of) Theorem 11. Similar to the construction in the underparametrized regime (for all $\beta$) and overparametrization regime (for $\beta = 0$), the distribution $\mathcal{D}$ can be made a product distribution.

**Theorem 11** (Overparametrized regime, $\beta$ being Gaussian). *Let $n < D - 9$. Given any sequence $\Delta_{n+8}, \Delta_{n+9}, \ldots, \Delta_{D-1}$ where $\Delta_d \in \{\uparrow, \downarrow\}$, there exists $\rho > 0$ and a distribution $\mathcal{D}$ such that for $\beta \sim \mathcal{N}(0, \rho^2)$ and every $n + 8 \leq d \leq D - 1$, we have*

$$L^{\text{exp}}_{d+1} \begin{cases} > L^{\text{exp}}_d, & \text{if } \Delta_d = \uparrow \\ < L^{\text{exp}}_d, & \text{if } \Delta_d = \downarrow. \end{cases}$$

*Proof of Theorem 11.* Define the design matrix $A_d \triangleq [x_1[1:d], \ldots, x_n[1:d]]^\top \in \mathbb{R}^{n \times d}$. Similar to the proof of Theorem 5, we construct the product distribution $\mathcal{D} = \prod_{d=1}^D \mathcal{D}_d$. We set $\mathcal{D}_d = \mathcal{N}(0, 1)$ for $d = 1, \ldots, n + 8$. For $n + 8 < d \leq D$, $\mathcal{D}_d$ is either $\mathcal{N}(0, \sigma_d^2)$ or $\mathcal{N}^{\text{mix}}_{\sigma_d, \mu_d}$ depending on $\Delta_d$ being either $\downarrow$ or $\uparrow$.

If $\Delta_{d-1} = \uparrow$, by Theorem 10, there exists $\sigma_d$ and $\mu_d$ such that $\mathcal{D}_d = \mathcal{N}^{\text{mix}}_{\sigma_d, \mu_d}$ guarantees $L^{\text{exp}}_d > L^{\text{exp}}_{d-1}$. If $\Delta_{d-1} = \downarrow$, define

$$\rho_d \triangleq \eta \sqrt{\frac{\mathbb{E}[\|(A_{d-1}^\top A_{d-1})^+ x_{\text{test}}[1:d-1]\|^2]}{\mathbb{E}\|A_{d-1}^{+\top} x_{\text{test}}[1:d-1]\|^2 + 1}}.$$

By Theorem 10, there exists $\sigma_d > 0$ such that if $\rho \leq \rho_d$ and $\mathcal{D}_d = \mathcal{N}(0, \sigma_d^2)$, then $L^{\text{exp}}_d < L^{\text{exp}}_{d-1}$. We take

$$\rho = \min_{d: \Delta_{d-1}=\downarrow} \rho_d.$$

$\square$

## 6 Conclusion

Our work proves that the expected risk of linear regression can manifest multiple descents when the number of features increases and sample size is fixed. This is carried out through an algorithmic construction of a feature-revealing process where the newly revealed feature follows either a Gaussian distribution or a Gaussian mixture distribution. Notably, the construction also enables us to control local maxima in the underparametrized regime and control ascents/descents freely in the overparametrized regime. Overall, this allows us to design the generalization curve away from the interpolation threshold.

We believe that our analysis of linear regression in this paper is a good starting point for explaining non-monotonic generalization curves observed in machine learning studies. Extending these results to more complex problem setups would be a meaningful future direction.

## Funding Transparency Statement

LC: Funding in direct support of this work: postdoctoral research fellowship by the Simons Institute for the Theory of Computing, University of California, Berkeley, and Google PhD Fellowship by Google. Additional revenues related to this work: internships at Google.

MB acknowledges support from NSF IIS-1815697, and the support of the NSF and the Simons Foundation for the Collaboration on the Theoretical Foundations of Deep Learning through awards DMS-2031883 and #814639.

AK: Funding in direct support of this work: NSF (IIS-1845032) and ONR (N00014-19-1-2406).

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
