## A  Almost Sure Convergence of Sequence of Normal Random Variables

In this paper, we need a sequence of random variables $\{X_n\}_{n\geq 1}$ such that $X_n \sim \mathcal{N}(0,\sigma_n^2)$, $\lim_{n\to+\infty} \sigma_n = 0$, and $X_n \to 0$ almost surely. The following lemma shows the existence of such a sequence.

**Lemma 12.** *There exist a sequence of random variables $\{X_n\}_{n\geq 1}$ such that $X_n \sim \mathcal{N}(0,\sigma_n^2)$, $\lim_{n\to+\infty} \sigma_n = 0$, and $X_n \to 0$ almost surely.*

*Proof.* Let $\sigma_n = 1/n^2$ and $X_n \sim \mathcal{N}(0,\sigma_n^2)$. Define the event $E_n \triangleq \{|X_n| > \varepsilon\}$. We have

$$\sum_{n=1}^{\infty} \mathbb{P}(E_n) = \sum_{n=1}^{\infty} \mathbb{P}(|\mathcal{N}(0,1)| > \varepsilon/\sigma_n) \leq \sum_{n=1}^{\infty} \frac{\sigma_n}{\varepsilon} e^{-\frac{\varepsilon^2}{2\sigma_n}} \leq \sum_{n=1}^{\infty} \frac{\sigma_n}{\varepsilon} = \sum_{n=1}^{\infty} \frac{1}{\varepsilon n^2} < +\infty\,.$$

By the Borel–Cantelli lemma, we have $\mathbb{P}(\limsup_{n\to+\infty} E_n) = 0$, which implies that $X_n \to 0$ almost surely. $\qquad\square$

## B  Proofs for Underparametrized Regime

### B.1  Proof of Lemma 2

By [7, Theorem 1], we have

$$\begin{bmatrix} A^\top \\ b^\top \end{bmatrix}^+ = \left[ (I-Q)A(A^\top(I-Q)A)^{-1}, \frac{(I-P)b}{b^\top(I-P)b} \cdot \right]$$

Define $r \triangleq A^\top b \in \mathbb{R}^d$. Since $A$ has linearly independent columns, the Gram matrix $G = A^\top A$ is non-singular. The Sherman-Morrison formula gives

$$(A^\top(I-Q)A)^{-1} = \left( A^\top A - \frac{rr^\top}{\|b\|^2} \right)^{-1} = G^{-1} + \frac{G^{-1}rr^\top G^{-1}}{\|b\|^2 - r^\top G^{-1}r} = G^{-1} + \frac{G^{-1}rb^\top(A^+)^\top}{\|b\|^2 - r^\top G^{-1}r}\,,$$

where we use the facts $r = A^\top b$ and $AG^{-1} = (A^+)^\top$ in the last equality. Therefore, we deduce

$$\begin{aligned}
A(A^\top(I-Q)A)^{-1} &= AG^{-1} + \frac{AG^{-1}rb^\top(A^+)^\top}{\|b\|^2 - r^\top G^{-1}r} \\
&= (A^+)^\top + \frac{AG^{-1}A^\top bb^\top(A^+)^\top}{\|b\|^2 - r^\top G^{-1}r} \\
&= \left( I + \frac{AA^+ bb^\top}{\|b\|^2 - r^\top G^{-1}r} \right)(A^+)^\top \\
&= \left( I + \frac{PQ}{1 - \frac{r^\top G^{-1}r}{\|b\|^2}} \right)(A^+)^\top\,.
\end{aligned}$$

Observe that

$$1 - \frac{r^\top G^{-1}r}{\|b\|^2} = 1 - \frac{b^\top A(A^\top A)^{-1}A^\top b}{\|b\|^2} = 1 - \frac{b^\top Pb}{\|b\|^2} = z\,.$$

Therefore, we obtain the desired expression.

### B.2  Proof of Theorem 3

First, we rewrite the expression as follows

$$\left\| \begin{bmatrix} A^\top \\ b^\top \end{bmatrix}^+ \begin{bmatrix} x \\ a_1 \end{bmatrix} \right\|^2 - \|(A^+)^\top x\|^2 \tag{6}$$

$$= \left\| (I-Q)(I + PQ/z)(A^+)^\top x + \frac{(I-P)b}{b^\top(I-P)b} a_1 \right\|^2 - \|(A^+)^\top x\|^2\,,$$

where $P, Q, z$ are defined in Lemma 2. Since $a_1$ has mean 0 and is independent of other random variables, so that the cross term vanishes under expectation over $b$ and $a_1$:

$$\mathbb{E}_{b,a_1}\left[\left\langle (I-Q)(I+PQ/z)(A^+)^\top x, \frac{(I-P)b}{b^\top(I-P)b}a_1\right\rangle\right] = 0\,,$$

where $\langle\cdot,\cdot\rangle$ denotes the inner product. Therefore taking the expectation of (6) over $b$ and $a_1$ yields

$$\mathbb{E}_{b,a_1}\left[\left\|\begin{bmatrix}A^\top\\b^\top\end{bmatrix}^+\begin{bmatrix}x\\a_1\end{bmatrix}\right\|^2 - \|(A^+)^\top x\|^2\right] \tag{7}$$

$$= \mathbb{E}_{b,a_1}\left[\|(I-Q)(I+PQ/z)(A^+)^\top x\|^2 - \|(A^+)^\top x\|^2 + \left\|\frac{(I-P)b}{b^\top(I-P)b}a_1\right\|^2\right] \tag{8}$$

$$\tag{9}$$

We simplify the third term. Recall that $I - P = I - AA^+$ is an orthogonal projection matrix and thus idempotent

$$\left\|\frac{(I-P)b}{b^\top(I-P)b}a_1\right\|^2 = \frac{a_1^2}{(b^\top(I-P)b)^2}\|(I-P)b\|^2 = \frac{a_1^2}{b^\top(I-P)b}\,. \tag{10}$$

Thus we have

$$\mathbb{E}_{b,a_1}\left[\left\|\begin{bmatrix}A^\top\\b^\top\end{bmatrix}^+\begin{bmatrix}x\\a_1\end{bmatrix}\right\|^2 - \|(A^+)^\top x\|^2\right] \tag{11}$$

$$= \mathbb{E}_{b,a_1}\left[\|(I-Q)(I+PQ/z)(A^+)^\top x\|^2 - \|(A^+)^\top x\|^2 + \frac{a_1^2}{b^\top(I-P)b}\right]\,. \tag{12}$$

We consider the first and second terms. We write $v = (A^+)^\top x$ and define $z = \frac{b^\top(I-P)b}{\|b\|^2}$. The sum of the first and second terms equals

$$\|(I-Q)(I+PQ/z)v\|^2 - \|v\|^2 = -v^\top M v\,, \tag{13}$$

where

$$M \triangleq Q - \frac{PQ+QP}{z} + \left(\frac{2}{z} - \frac{1}{z^2}\right)QPQ + \frac{QPQPQ}{z^2}\,.$$

The rank of $M$ is at most 2. To see this, we re-write $M$ in the following way

$$M = \left[Q\left(-\frac{P}{z} + \left(\frac{2}{z} - \frac{1}{z^2}\right)PQ + \frac{PQPQ}{z^2}\right)\right] + \left[-\frac{PQ}{z}\right] \triangleq M_1 + M_2\,.$$

Notice that $\operatorname{rank}(M_1) \leq \operatorname{rank}(Q)$, $\operatorname{rank}(M_2) \leq \operatorname{rank}(Q)$, and $\operatorname{rank}(Q) = 1$.

It follows that $\operatorname{rank}(M) \leq \operatorname{rank}(M_1) + \operatorname{rank}(M_2) = 2$. The matrix $M$ has at least $n - 2$ zero eigenvalues. We claim that $M$ has two non-zero eigenvalues and they are $1 - 1/z < 0$ and $1$.

Since

$$\operatorname{rank}(PQ) \leq \operatorname{rank}(Q) = 1$$

and

$$\operatorname{tr}(PQ) = \frac{b^\top Pb}{\|b\|^2} = 1 - z,$$

thus $PQ$ has a unique non-zero eigenvalue $1 - z$. Let $u \neq 0$ denote the corresponding eigenvector such that $PQu = (1-z)u$. Since $u \in \operatorname{im} P$ and $P$ is a projection, we have $Pu = u$. Therefore we can verify that

$$Mu = (1 - \frac{1}{z})u\,.$$

To show that the other non-zero eigenvalue of $M$ is 1, we compute the trace of $M$

$$\operatorname{tr}(M) = \operatorname{tr}(Q) - \frac{2\operatorname{tr}(PQ)}{z} + \left(\frac{2}{z} - \frac{1}{z^2}\right)\operatorname{tr}(PQ) + \frac{\operatorname{tr}((PQ)^2)}{z^2} = 2 - \frac{1}{z},$$

where we use the fact that $\operatorname{tr}(Q) = 1$, $\operatorname{tr}(PQ) = 1 - z$,

$$\operatorname{tr}((PQ)^2) = \operatorname{tr}\left(\frac{Pbb^\top Pbb^\top}{\|b\|^4}\right) = \operatorname{tr}\left(\frac{(b^\top Pb)(b^\top Pb)}{\|b\|^4}\right) = (1-z)^2.$$

We have shown that $M$ has eigenvalue $1 - 1/z$ and $M$ has at most two non-zero eigenvalues. Therefore, the other non-zero eigenvalue is $\operatorname{tr}(M) - (1 - 1/z) = 1$.

We are now in a position to upper bound (13) as follows:

$$-v^\top M v \leq -(1 - 1/z)\|v\|^2.$$

Putting all three terms of the change in the dimension-normalized generalization loss yields

$$\mathbb{E}_{b,a_1}\left[\left\|\begin{bmatrix}A^\top\\b^\top\end{bmatrix}^+\begin{bmatrix}x\\a_1\end{bmatrix}\right\|^2 - \left\|(A^+)^\top x\right\|^2\right] \leq \mathbb{E}_{b,a_1}\left[-(1-1/z)\|v\|^2 + \frac{a_1^2}{b^\top(I-P)b}\right].$$

Therefore, we get

$$\mathbb{E}_{b,a_1}\left[\left\|\begin{bmatrix}A^\top\\b^\top\end{bmatrix}^+\begin{bmatrix}x\\a_1\end{bmatrix}\right\|^2\right] \leq \mathbb{E}_{b,a_1}\left[\frac{1}{z}\|v\|^2 + \frac{a_1^2}{b^\top(I-P)b}\right].$$

For $b_1, \ldots, b_n, a_1 \overset{iid}{\sim} \mathcal{N}(0,1)$, we have $\mathbb{E}[a_1^2] = 1$. Moreover, $b^\top(I - P)b$ follows $\chi^2(n - d)$ a distribution. Thus $\frac{1}{b^\top(I-P)b}$ follows an inverse-chi-squared distribution with mean $\frac{1}{n-d-2}$. Therefore the expectation $\mathbb{E}[\frac{a_1^2}{b^\top(I-P)b}] = \frac{1}{n-d-2}$.

Notice that $1/z$ follows a $1 + \frac{d}{n-d}F(d, n-d)$ distribution and thus $\mathbb{E}[1/z] = 1 + \frac{d}{n-d-2}$.

As a result, we obtain

$$\mathbb{E}_{b,a_1}\left[\left\|\begin{bmatrix}A^\top\\b^\top\end{bmatrix}^+\begin{bmatrix}x\\a_1\end{bmatrix}\right\|^2\right] \leq \frac{(n-2)\|v\|^2 + 1}{n-d-2}$$

For $b_1, \ldots, b_n, a_1 \overset{iid}{\sim} \mathcal{N}_{\sigma,1}^{\mathrm{mix}}$, we need the following lemma.

**Lemma 13** (Proof in Appendix B.3). *Assume $d$, $n > d + 2$ and $P$ are fixed, where $P \in \mathbb{R}^{n\times n}$ is an orthogonal projection matrix whose rank is $d$. Define $z \triangleq \frac{b^\top(I-P)b}{\|b\|^2}$, where $b = [b_1, \ldots, b_n]^\top \in \mathbb{R}^n$. If $a_1, b_1, \cdots, b_n \overset{iid}{\sim} \mathcal{N}_{\sigma,1}^{\mathrm{mix}}$, we have $\mathbb{E}[1/z] \leq \frac{n-2+\sqrt{d}}{n-d-2}$ and $\mathbb{E}[a_1^2/b^\top(I-P)b] \leq \frac{2/(3\sigma^2)+1}{n-d-2}$.*

Lemma 13 implies that

$$\mathbb{E}_{b,a_1}[1/z] \leq \frac{n-2+\sqrt{d}}{n-d-2}, \quad \mathbb{E}_{b,a_1}\left[\frac{a_1^2}{b^\top(I-P)b}\right] < \frac{2/(3\sigma^2)+1}{n-d-2}.$$

Therefore, we conclude that

$$\mathbb{E}_{b,a_1}\left\|\begin{bmatrix}A^\top\\b^\top\end{bmatrix}^+\begin{bmatrix}x\\a_1\end{bmatrix}\right\|^2 \leq \frac{(n-2+\sqrt{d})\|v\|^2 + 2/(3\sigma^2)+1}{n-d-2}.$$

## B.3 Proof of Lemma 13

Lemma 14 shows that a noncentral $\chi^2$ distribution first-order stochastically dominates a central $\chi^2$ distribution of the same degree of freedom. It will be needed in the proof of Lemma 13.

**Lemma 14.** *Assume that random variables $X \sim \chi^2(k, \lambda)$ and $Y \sim \chi^2(k)$, where $\lambda > 0$. For any $c > 0$, we have*

$$\mathbb{P}(X \geq c) > \mathbb{P}(Y \geq c).$$

*In other words, the random variable $X$ (first-order) stochastically dominates $Y$.*

*Proof.* Let $Y_1, X_2, \ldots, X_k \overset{iid}{\sim} \mathcal{N}(0, 1)$ and $X_1 \sim \mathcal{N}(\sqrt{\lambda}, 1)$ and all these random variables are jointly independent. Then $X' \triangleq \sum_{i=1}^{k} X_i^2 \sim \chi^2(k, \lambda)$ and $Y' \triangleq Y_1^2 + \sum_{i=2}^{k} X_i^2 \sim \chi^2(k)$.

It suffices to show that $\mathbb{P}(X' \geq c) > \mathbb{P}(Y' \geq c)$, or equivalently, $\mathbb{P}(|\mathcal{N}(\mu, 1)| \geq c) > \mathbb{P}(|\mathcal{N}(0, 1)| \geq c)$ for all $c > 0$ and $\mu \triangleq \sqrt{\lambda} > 0$. Denote $F_c(t) = \mathbb{P}(|\mathcal{N}(\mu, 1)| \geq c)$ and we have

$$F_c(\mu) = 1 - \frac{1}{\sqrt{2\pi}} \int_{-c}^{c} \exp\left(-\frac{(x - \mu)^2}{2}\right) dx = 1 - \frac{1}{\sqrt{2\pi}} \int_{-c-\mu}^{c-\mu} \exp\left(-\frac{x^2}{2}\right) dx,$$

and thus

$$\frac{dF_c(\mu)}{d\mu} = \frac{1}{\sqrt{2\pi}} \left[\exp\left(-\frac{(c - \mu)^2}{2}\right) - \exp\left(-\frac{(c + \mu)^2}{2}\right)\right] > 0.$$

This shows $\mathbb{P}(|\mathcal{N}(\mu, 1)| \geq c) > \mathbb{P}(|\mathcal{N}(0, 1)| \geq c)$ and we are done.

$\square$

*Proof of Lemma 13.* Since $b_i \overset{iid}{\sim} \mathcal{N}_{\sigma,1}^{\mathrm{mix}}$, we can rewrite $b = u + w$ where $w \sim \mathcal{N}(0, \sigma^2 I_n)$ and the entries of $u$ satisfy $u_i \overset{iid}{\sim} \mathrm{Unif}(\{-1, 0, 1\})$. Furthermore, $u$ and $w$ are independent. Similarly, we can write $a_1 = \hat{u} + \hat{w}$, where $\hat{u} \sim \mathrm{Unif}(\{-1, 0, 1\})$ and $\hat{w} \sim \mathcal{N}(0, \sigma^2)$ are independent. To bound $\mathbb{E}[a_1^2]$, we have

$$\mathbb{E}[a_1^2] = \mathbb{E}[(\hat{u} + \hat{w})^2] = \mathbb{E}[\hat{u}^2] + \mathbb{E}[\hat{w}^2] = \frac{2}{3} + \sigma^2.$$

Note that

$$\frac{1}{z} = \frac{b^\top I b}{b^\top (I - P) b} = 1 + \frac{(u + w)^\top P (u + w)}{(u + w)^\top (I - P)(u + w)}.$$

Since $P$ is an orthogonal projection, there exists an orthogonal transformation $O$ depending only on $P$ such that

$$(u + w)^\top P (u + w) = [O(u + w)]^\top D_d [O(u + w)]$$

where $D_d = \mathrm{diag}([1, \ldots, 1, 0 \ldots, 0])$ with $d$ diagonal entries equal to 1 and the others equal to 0. We denote $\tilde{u} = O(u)$, which is fixed (as $u$ and $O$ are fixed), and $\tilde{w} = O(w) \sim \mathcal{N}(0, \sigma^2 I_n)$. It follows that

$$\frac{1}{z} = 1 + \frac{(\tilde{u} + \tilde{w})^\top D_d (\tilde{u} + \tilde{w})}{(\tilde{u} + \tilde{w})^\top (I - D_d)(\tilde{u} + \tilde{w})} = 1 + \frac{\sum_{i=1}^{d} (\tilde{u}_i + \tilde{w}_i)^2}{\sum_{i=d+1}^{n} (\tilde{u}_i + \tilde{w}_i)^2} = 1 + \frac{\sum_{i=1}^{d} (\tilde{u}_i + \tilde{w}_i)^2/\sigma^2}{\sum_{i=d+1}^{n} (\tilde{u}_i + \tilde{w}_i)^2/\sigma^2}.$$

Observe that

$$\sum_{i=1}^{d} (\tilde{u}_i + \tilde{w}_i)^2/\sigma^2 \sim \chi^2\left(d, \sqrt{\sum_{i=1}^{d} \tilde{u}_i^2}\right)$$

$$\sum_{i=d+1}^{n} (\tilde{u}_i + \tilde{w}_i)^2/\sigma^2 \sim \chi^2\left(n - d, \sqrt{\sum_{i=d+1}^{n} \tilde{u}_i^2}\right),$$

and that these two quantities are independent. It follows that

$$\mathbb{E}\left[\sum_{i=1}^{d} (\tilde{u}_i + \tilde{w}_i)^2/\sigma^2 \,\middle|\, u\right] = d + \sqrt{\sum_{i=1}^{d} \tilde{u}_i^2}.$$

By Lemma 14, the denominator $\sum_{i=d+1}^{n}(\tilde{u}_i + \tilde{w}_i)^2/\sigma^2$ first-order stochastically dominates $\chi^2(n-d)$. Therefore, we have

$$\mathbb{E}\left[\frac{1}{\sum_{i=d+1}^{n}(\tilde{u}_i + \tilde{w}_i)^2/\sigma^2}\bigg|u\right] \le \mathbb{E}\left[\frac{1}{\chi^2(n-d)}\right] = \frac{1}{n-d-2}.$$

Putting the numerator and denominator together yields

$$\mathbb{E}\left[\frac{1}{z}\bigg|u\right] \le 1 + \frac{d + \sqrt{\sum_{i=1}^{d}\tilde{u}_i^2}}{n-d-2} \le 1 + \frac{d+\sqrt{d}}{n-d-2} = \frac{n-2+\sqrt{d}}{n-d-2}.$$

Similarly, we have

$$\mathbb{E}\left[\frac{1}{b^\top(I-P)b}\bigg|u\right] = \mathbb{E}\left[\frac{1}{[O(u+w)]^\top(I-D_d)[O(u+w)]}\bigg|u\right]$$

$$= \mathbb{E}\left[\frac{1/\sigma^2}{\sum_{i=d+1}^{n}(\tilde{u}_i+\tilde{w}_i)^2/\sigma^2}\bigg|u\right]$$

$$\le \frac{1}{\sigma^2}\mathbb{E}\left[\frac{1}{\chi^2(n-d)}\right]$$

$$= \frac{1}{\sigma^2}\cdot\frac{1}{n-d-2}.$$

Thus, we obtain

$$\mathbb{E}[1/z] \le \frac{n-2+\sqrt{d}}{n-d-2}, \quad \mathbb{E}\left[\frac{1}{b^\top(I-P)b}\right] \le \frac{1}{\sigma^2}\cdot\frac{1}{n-d-2}.$$

It follows that

$$\mathbb{E}\left[\frac{a_1^2}{b^\top(I-P)b}\right] \le \frac{2/3+\sigma^2}{\sigma^2}\cdot\frac{1}{n-d-2} = \frac{2/(3\sigma^2)+1}{n-d-2}.$$

$\square$

## B.4 Proof of Theorem 4

We start from (12). Taking expectation over all random variables gives

$$\mathbb{E}\left[\left\|\begin{bmatrix}A^\top\\b^\top\end{bmatrix}^+\begin{bmatrix}x\\a_1\end{bmatrix}\right\|^2 - \|(A^+)^\top x\|^2\right]$$

$$= \mathbb{E}\left[\|(I-Q)(I+PQ/z)(A^+)^\top x\|^2 - \|(A^+)^\top x\|^2 + \frac{a_1^2}{b^\top(I-P)b}\right]$$

$$\ge -\mathbb{E}\|(A^+)^\top x\|^2 + \mathbb{E}\left[\frac{a_1^2}{\sum_{i=1}^{n}b_i^2}\right].$$

Our strategy is to choose $\sigma$ so that $\mathbb{E}\left[\frac{a_1^2}{\sum_{i=1}^{n}b_i^2}\right]$ is sufficiently large. This is indeed possible as we immediately show. Define independent random variables $u \sim \text{Unif}(\{-1,0,1\})$ and $w \sim \mathcal{N}(0,\sigma^2)$. Since $a_1$ has the same distribution as $u+w$, we have

$$\mathbb{E}[a_1^2] = \mathbb{E}[(u+w)^2] = \mathbb{E}[u^2] + \mathbb{E}[w^2] \ge \frac{2}{3}.$$

On the other hand,

$$\mathbb{E}\left[\frac{1}{\sum_{i=1}^{n}b_i^2}\right] \ge \mathbb{P}(\max_i|b_i|\le\sigma)\,\mathbb{E}\left[\frac{1}{\sum_{i=1}^{n}b_i^2}\bigg|\max_i|b_i|\le\sigma\right]$$

$$= [\mathbb{P}(|b_1|\le\sigma)]^n\,\mathbb{E}\left[\frac{1}{\sum_{i=1}^{n}b_i^2}\bigg|\max_i|b_i|\le\sigma\right]$$

$$\ge \left[\frac{1}{3\sqrt{2\pi\sigma^2}}\int_{-\sigma}^{\sigma}\exp\left(-\frac{t^2}{2\sigma^2}\right)dt\right]^n\frac{1}{n\sigma^2}$$

$$\ge \frac{1}{5^n n\sigma^2}.$$

Together we have
$$\mathbb{E}\left[\frac{a_1^2}{\sum_{i=1}^n b_i^2}\right] \geq \frac{1}{5^{n+1}n\sigma^2}\,.$$

As a result, we conclude
$$\lim_{\sigma\to 0^+}\mathbb{E}\left[\left\|\begin{bmatrix}A^\top\\b^\top\end{bmatrix}^+\begin{bmatrix}x\\a_1\end{bmatrix}\right\|^2 - \|(A^+)^\top x\|^2\right] = +\infty\,,$$

which completes the proof.

## C  Proofs for Overparametrized Regime

### C.1  Proof of Lemma 6

Since $A$ and $B$ have full row rank, $(AA^\top)^{-1}$ and $(BB^\top)^{-1}$ exist. Therefore we have
$$B^+ = B^\top(BB^\top)^{-1}.$$

The Sherman-Morrison formula gives
$$(BB^\top)^{-1} = (AA^\top + bb^\top)^{-1} = G - \frac{Gbb^\top G}{1 + b^\top Gb} = G - Gbu = G(I - bu)\,.$$

Hence, we deduce
$$B^+ = [A, b]^\top G(I - bu) = \begin{bmatrix}A^\top G(I - bu)\\b^\top G(I - bu)\end{bmatrix} = \begin{bmatrix}A^+(I - bu)\\b^\top G(I - bu)\end{bmatrix} = \begin{bmatrix}A^+(I - bu)\\u\end{bmatrix}\,.$$

Transposing the above equation yields to the promised equation.

### C.2  Proof of Lemma 7

Let us first denote
$$v \triangleq (A^+)^\top x$$
and
$$G \triangleq (AA^\top)^{-1} \in \mathbb{R}^{n\times n}.$$
First note that by Cauchy-Schwarz inequality, it suffices to show there exists $\mathcal{D}$ such that $\mathbb{E}[\lambda_{\max}^4(G)] < +\infty$ and $\mathbb{E}\|v\|^4 < +\infty$.

We define $A_d \in \mathbb{R}^{n\times d}$ to be the submatrix of $A$ that consists of all $n$ rows and first $d$ columns. Denote
$$G_d \triangleq (A_dA_d^\top)^{-1} \in \mathbb{R}^{n\times n}.$$
We will prove $\mathbb{E}[\lambda_{\max}^4(G)] < +\infty$ by induction.

The base step is $d = n + 8$. Recall $\mathcal{D}_{[1:n+8]} = \mathcal{N}(0, I_{n+8})$. We first show $\mathbb{E}[\lambda_{\max}(G_{n+8})]^4 < +\infty$. Note that since $G_{n+8}$ is almost surely positive definite,
$$\mathbb{E}[\lambda_{\max}^4(G_{n+8})] = \mathbb{E}[\lambda_{\max}(G_{n+8}^4)] \leq \mathbb{E}\operatorname{tr}(G_{n+8}^4) = \mathbb{E}\operatorname{tr}((A_{n+8}A_{n+8}^\top)^{-4}) = \operatorname{tr}(\mathbb{E}[(A_{n+8}A_{n+8}^\top)^{-4}])\,.$$

By our choice of $\mathcal{D}_{[1:n+8]}$, the matrix $(A_{n+8}A_{n+8}^\top)^{-1}$ is an inverse Wishart matrix of size $n \times n$ with $(n + 8)$ degrees of freedom, and thus has finite fourth moment (see, for example, Theorem 4.1 in [57]). It then follows that
$$\mathbb{E}[\lambda_{\max}^4(G_{n+8})] \leq \operatorname{tr}(\mathbb{E}[(A_{n+8}A_{n+8}^\top)^{-4}]) < +\infty\,.$$

For the inductive step, assume $\mathbb{E}[\lambda_{\max}(G_d)]^4 < +\infty$ for some $d \geq n + 8$. We claim that
$$\lambda_{\max}(G_{d+1}) \leq \lambda_{\max}(G_d)\,,$$
or equivalently,
$$\lambda_{\min}(A_dA_d^\top) \leq \lambda_{\min}(A_{d+1}A_{d+1}^\top)\,.$$

Indeed, this follows from the fact that

$$A_d A_d^\top \preccurlyeq A_d A_d^\top + bb^\top = A_{d+1} A_{d+1}^\top,$$

under the Loewner order, where $b \in \mathbb{R}^{n \times 1}$ is the $(d+1)$-th column of $A$. Therefore, we have

$$\mathbb{E}[\lambda_{\max}^4(G_{d+1})] \leq \mathbb{E}[\lambda_{\max}^4(G_d)]$$

and by induction, we conclude that $\mathbb{E}[\lambda_{\max}^4(G)] < +\infty$ for all $d \geq n + 8$.

Now we proceed to show $\mathbb{E}\|v\|^4 < +\infty$. We have

$$\|v\|^4 = \|(AA^\top)^{-1}Ax\|^4 \leq \|(AA^\top)^{-1}A\|_{op}^4 \cdot \|x\|^4,$$

where $\|\cdot\|_{op}$ denotes the $\ell^2 \to \ell^2$ operator norm. Note that

$$\begin{aligned}
\|(AA^\top)^{-1}A\|_{op}^4 &= \lambda_{\max}^2 \left( \left((AA^\top)^{-1}A\right)^\top (AA^\top)^{-1}A \right) \\
&= \lambda_{\max}^2 \left(A^\top(AA^\top)^{-2}A\right) \\
&= \lambda_{\max} \left( \left(A^\top(AA^\top)^{-2}A\right)^2 \right),
\end{aligned}$$

where the last equality uses the fact that $A^\top(AA^\top)^{-2}A$ is positive semidefinite. Moreover, we deduce

$$\begin{aligned}
\|(AA^\top)^{-1}A\|_{op}^4 &= \lambda_{\max}\left(A^\top(AA^\top)^{-3}A\right) \\
&\leq \operatorname{tr}\left(A^\top(AA^\top)^{-3}A\right) \\
&= \operatorname{tr}\left((AA^\top)^{-3}AA^\top\right) \\
&= \operatorname{tr}\left((AA^\top)^{-2}\right).
\end{aligned}$$

Using the fact that $A_d A_d^\top \preccurlyeq A_{d+1} A_{d+1}^\top$ established above, induction gives

$$(AA^\top)^{-2} \preccurlyeq (A_{n+8} A_{n+8}^\top)^{-2}.$$

It follows that

$$\mathbb{E}\left[\|(AA^\top)^{-1}A\|_{op}^4\right] \leq \mathbb{E}\left[\operatorname{tr}\left(\left(A_{n+8}A_{n+8}^\top\right)^{-2}\right)\right] = \operatorname{tr}\left(\mathbb{E}\left[\left(A_{n+8}A_{n+8}^\top\right)^{-2}\right]\right) < +\infty, \quad (14)$$

where again we use that fact that inverse Wishart matrix $\left(A_{n+8}A_{n+8}^\top\right)^{-1}$ has finite second moment.

Next, we demonstrate $\mathbb{E}\|x\|^4 < +\infty$. Recall that every $\mathcal{D}_i$ is either a Gaussian or a Gaussian mixture distribution. Therefore, every entry of $x$ has a subgaussian tail, and thus $\mathbb{E}\|x\|^4 < +\infty$. Together with (14) and the fact that $x$ and $A$ are independent, we conclude that

$$\mathbb{E}\|v\|^4 \leq \mathbb{E}\left[\|(AA^\top)^{-1}A\|_{op}^4\right] \cdot \mathbb{E}\left[\|x\|^4\right] < +\infty.$$

### C.3 Proof of Theorem 8

The randomness comes from $A, x, a_1$ and $b$. We first condition on $A$ and $x$ being fixed.

Let $G \triangleq (AA^\top)^{-1} \in \mathbb{R}^{n \times n}$ and $u \triangleq \frac{b^\top G}{1+b^\top Gb} \in \mathbb{R}^{1 \times n}$. Define

$$v \triangleq (A^+)^\top x, \quad r \triangleq 1 + b^\top Gb, \quad H \triangleq bb^\top.$$

We compute the left-hand side but take the expectation over only $a_1$ for the moment

$$\begin{aligned}
\mathbb{E}_y &\left\| \begin{bmatrix} A^\top \\ b^\top \end{bmatrix}^+ \begin{bmatrix} x \\ a_1 \end{bmatrix} \right\|^2 - \|(A^+)^\top x\|^2 \\
&= \mathbb{E}_y \left\| (I - bu)^\top v + u^\top a_1 \right\|^2 - \|v\|^2 \\
&= \|(I - bu)^\top v\|^2 + \mathbb{E}_y \|u^\top a_1\|^2 - \|v\|^2 && (\mathbb{E}[a_1] = 0) \\
&= \|(I - bu)^\top v\|^2 + \mathbb{E}_y[a_1^2] \frac{\|Gb\|^2}{r^2} - \|v\|^2.
\end{aligned}$$

Let us first consider the first and third terms of the above equation:

$$\|(I - bu)^\top v\|^2 - \|v\|^2 = v^\top \left((I - bu)(I - bu)^\top - I\right) v$$
$$= -v^\top \left(bu + u^\top b^\top - buu^\top b^\top\right) v$$
$$= -v^\top \left(\frac{HG + GH}{r} - \frac{HG^2H}{r^2}\right) v \,.$$

Write $G = V\Lambda V^\top$, where $\Lambda = \mathrm{diag}(\lambda_1, \ldots, \lambda_n) \in \mathbb{R}^{n \times n}$ is a diagonal matrix ($\lambda_i > 0$) and $V \in \mathbb{R}^{n \times n}$ is an orthogonal matrix. Recall $b \sim \mathcal{N}(0, \sigma^2 I_n)$. Therefore $w \triangleq V^\top b \sim \mathcal{N}(0, \sigma^2 I_n)$. Taking the expectation over $b$, we have

$$\mathbb{E}_b\left[\frac{HG + GH}{r}\right] = \mathbb{E}_b\left[V\frac{V^\top bb^\top V\Lambda + \Lambda V^\top bb^\top V}{1 + b^\top V\Lambda V^\top b}V^\top\right] = V\mathbb{E}_w\left[\frac{ww^\top\Lambda + \Lambda ww^\top}{1 + w^\top\Lambda w}\right]V^\top \,.$$

Let $R \triangleq \mathbb{E}_w\left[\frac{ww^\top\Lambda + \Lambda ww^\top}{1 + w^\top\Lambda w}\right]$. We have

$$R_{ii} = \mathbb{E}_w\left[\frac{2\lambda_i w_i^2}{1 + \sum_{i=1}^n \lambda_i w_i^2}\right] = \sigma^2 \mathbb{E}_{\nu \sim \mathcal{N}(0, I_n)}\left[\frac{2\lambda_i \nu_i^2}{1 + \sigma^2 \sum_{i=1}^n \lambda_i \nu_i^2}\right] > 0$$

and if $i \neq j$,

$$R_{ij} = \mathbb{E}_w\left[\frac{(\lambda_i + \lambda_j)w_i w_j}{1 + \sum_{i=1}^n \lambda_i w_i^2}\right] \,.$$

Notice that for any $w$ and $j$, it has the same distribution if we replace $w_j$ by $-w_j$. As a result,

$$R_{ij} = \mathbb{E}_w\left[\frac{(\lambda_i + \lambda_j)w_i(-w_j)}{1 + \sum_{i=1}^n \lambda_i w_i^2}\right] = -R_{ij} \,.$$

Thus the matrix $R$ is a diagonal matrix and

$$R = 2\sigma^2 \frac{\Lambda \mathrm{diag}(\nu)^2}{1 + \sigma^2 \nu^\top \Lambda \nu} \,.$$

Thus we get

$$\mathbb{E}_{b,A}\left[\frac{HG + GH}{r}\right] = 2\sigma^2 \mathbb{E}_{\nu \sim \mathcal{N}(0, I_n), A}\left[\frac{GV \mathrm{diag}(\nu)^2 V^\top}{1 + \sigma^2 \nu^\top \Lambda \nu}\right]$$

Moreover, by the monotone convergence theorem, we deduce

$$\lim_{\sigma \to 0^+} \mathbb{E}_{\nu \sim \mathcal{N}(0, I_n), A, x}\left[-v^\top \frac{GV \mathrm{diag}(\nu)^2 V^\top}{1 + \sigma^2 \nu^\top \Lambda \nu} v\right] = \mathbb{E}_{\nu \sim \mathcal{N}(0, I_n), A, x}\left[-v^\top GV \mathrm{diag}(\nu)^2 V^\top v\right]$$
$$= \mathbb{E}[-v^\top Gv] \,.$$

It follows that as $\sigma \to 0^+$,

$$\mathbb{E}\left[-v^\top \frac{HG + GH}{r} v\right] \sim -2\sigma^2 \mathbb{E}[v^\top Gv] = -2\sigma^2 \mathbb{E}\left[v^\top (AA^\top)^{-1} v\right] = -2\sigma^2 \mathbb{E}[\|(A^\top A)^+ x\|^2] \,.$$

Moreover, by (4), we have

$$\mathbb{E}\left[v^\top (AA^\top)^{-1} v\right] \leq \mathbb{E}\left[\lambda_{\max}\left((AA^\top)^{-1}\right)\|(A^+)^\top x\|^2\right] < +\infty \,.$$

Next, we study the term $HG^2H/r^2$:

$$\mathbb{E}_{b,A}\left[\frac{HG^2H}{r^2}\right] = \mathbb{E}_{b,A}\left[V\frac{V^\top bb^\top V\Lambda^2 V^\top bb^\top V}{(1 + b^\top V\Lambda V^\top b)^2}V^\top\right]$$
$$= \mathbb{E}_{w \sim \mathcal{N}(0, \sigma^2 I_n), A}\left[V\frac{ww^\top\Lambda^2 ww^\top}{(1 + w^\top\Lambda w)^2}V^\top\right]$$
$$= \sigma^4 \mathbb{E}_{\nu \sim \mathcal{N}(0, I_n), A}\left[V\frac{\nu\nu^\top\Lambda^2\nu\nu^\top}{(1 + \sigma^2 \nu^\top\Lambda\nu)^2}V^\top\right] \,.$$

Again, by the monotone convergence theorem, we have

$$\lim_{\sigma \to 0^+} \mathbb{E}_{\nu \sim \mathcal{N}(0,I_n),A,x} \left[ v^\top V \frac{\nu\nu^\top \Lambda^2 \nu\nu^\top}{(1 + \sigma^2 \nu^\top \Lambda \nu)^2} V^\top v \right]$$

$$= \mathbb{E}_{\nu \sim \mathcal{N}(0,I_n),A,x} \left[ v^\top V \nu\nu^\top \Lambda^2 \nu\nu^\top V^\top v \right]$$

$$= \mathbb{E}_{A,x} \left[ v^\top V \left( 2\Lambda^2 + I_n \sum_{i=1}^n \lambda_i^2 \right) V^\top v \right]$$

$$= \mathbb{E} \left[ v^\top \left( 2G^2 + \operatorname{tr}(G^2)I_n \right) v \right] .$$

It follows that as $\sigma \to 0^+$,

$$\mathbb{E}_{b,A,x} \left[ \frac{HG^2 H}{r^2} \right]$$

$$\sim \sigma^4 \mathbb{E} \left[ v^\top \left( 2G^2 + \operatorname{tr}(G^2)I_n \right) v \right]$$

$$= \sigma^4 \mathbb{E} \left[ 2\|(AA^\top)^{-1}v\|^2 + \operatorname{tr}((AA^\top)^{-2})\|v\|^2 \right] .$$

Moreover, by (4), we have

$$\mathbb{E} \left[ 2\|(AA^\top)^{-1}v\|^2 + \operatorname{tr}((AA^\top)^{-2})\|v\|^2 \right] \le (n+2)\mathbb{E} \left[ \lambda_{\max}^2((AA^\top)^{-1})\|(A^+)^\top x\|^2 \right] < +\infty .$$

We apply a similar method to the term $\frac{\|Gb\|^2}{r^2}$. We deduce

$$\frac{\|Gb\|^2}{r^2} = \frac{b^\top G^2 b}{(1 + b^\top Gb)^2} = \frac{b^\top V \Lambda^2 V^\top b}{(1 + b^\top V \Lambda V^\top b)^2} .$$

It follows that

$$\mathbb{E} \left[ \frac{\|Gb\|^2}{r^2} \right] = \mathbb{E}_{w \sim \mathcal{N}(0,\sigma^2 I_n),A} \left[ \frac{w^\top \Lambda^2 w}{(1 + w^\top \Lambda w)^2} \right] = \sigma^2 \mathbb{E}_{\nu \sim \mathcal{N}(0,I_n),A} \left[ \frac{\nu^\top \Lambda^2 \nu}{(1 + \sigma^2 \nu^\top \Lambda \nu)^2} \right]$$

The monotone convergence theorem implies

$$\lim_{\sigma \to 0^+} \mathbb{E}_{\nu \sim \mathcal{N}(0,I_n),A} \left[ \frac{\nu^\top \Lambda^2 \nu}{(1 + \sigma^2 \nu^\top \Lambda \nu)^2} \right] = \mathbb{E}[\nu^\top \Lambda^2 \nu] = \mathbb{E}[\operatorname{tr}(G^2)] .$$

Thus we get as $\sigma \to 0^+$

$$\mathbb{E}_y[a_1^2] \frac{\|Gb\|^2}{r^2} \sim \sigma^4 \mathbb{E}[\operatorname{tr}(G^2)] ,$$

where $\mathbb{E}[\operatorname{tr}(G^2)] \le n\mathbb{E}[\lambda_{\max}^2((AA^\top)^{-1})] < +\infty$.

Putting all three terms together, we have as $\sigma \to 0^+$

$$L_{d+1} - L_d \sim -2\sigma^2 \mathbb{E}[\|(A^\top A)^+ x\|^2] .$$

Therefore, there exists $\sigma > 0$ such that $L_{d+1} - L_d < 0$.

## C.4    Proof of Theorem 9

Again we first condition on $A$ and $x$ being fixed. Let $G \triangleq (AA^\top)^{-1} \in \mathbb{R}^{n \times n}$ and $u \triangleq \frac{b^\top G}{1 + b^\top Gb} \in \mathbb{R}^{1 \times n}$ as defined in Lemma 6. We also define the following variables:

$$v \triangleq (A^+)^\top x , \quad r \triangleq 1 + b^\top Gb .$$

We compute $L_{d+1} - L_d$ but take the expectation over only $a_1$ for the moment

$$\mathbb{E}_y \left\| \begin{bmatrix} A^\top \\ b^\top \end{bmatrix}^+ \begin{bmatrix} x \\ a_1 \end{bmatrix} \right\|^2 - \|(A^+)^\top x\|^2$$

$$= \mathbb{E}_y \left\| (I - bu)^\top v + u^\top a_1 \right\|^2 - \|v\|^2$$

$$= \|(I - bu)^\top v\|^2 + \mathbb{E}_y \|u^\top a_1\|^2 - \|v\|^2 \qquad (\mathbb{E}[a_1] = 0)$$

$$= \|(I - bu)^\top v\|^2 + \mathbb{E}_y[a_1^2] \frac{\|Gb\|^2}{r^2} - \|v\|^2 . \qquad (15)$$

Our strategy is to make $\mathbb{E}[a_1^2 \frac{\|Gb\|^2}{r^2}]$ arbitrarily large. To this end, by the independence of $a_1$ and $b$ we have

$$\mathbb{E}_{a_1,b}\left[a_1^2 \frac{\|Gb\|^2}{r^2}\right] = \mathbb{E}_y[a_1^2]\mathbb{E}_b\left[\frac{\|Gb\|^2}{r^2}\right].$$

By definition of $\mathcal{N}_{\sigma,\mu}^{\mathrm{mix}}$, with probability $2/3$, $a_1$ is sampled from either $\mathcal{N}(\mu,\sigma^2)$ or $\mathcal{N}(-\mu,\sigma^2)$, which implies $\mathbb{E}[a_1^2] \geq \frac{1}{3}\mu^2$. For each $b_i$, we have

$$\mathbb{P}(|b_i| \in [\sigma, 2\sigma]) \geq \frac{1}{3} \times \frac{1}{4}.$$

Also note that $G$ is positive definite. It follows that

$$\mathbb{E}_b\left[\frac{\|Gb\|^2}{r^2}\right] = \mathbb{E}_b\left[\frac{\|Gb\|^2}{(1+b^\top Gb)^2}\right] \geq \mathbb{E}_b\left[\frac{(\lambda_{\min}(G)\|b\|)^2}{(1+\lambda_{\max}(G)\|b\|^2)^2}\right] \geq \left(\frac{1}{12}\right)^n \frac{\lambda_{\min}^2(G)n\sigma^2}{(1+4\lambda_{\max}(G)n\sigma^2)^2}.$$

Altogether we have

$$\mathbb{E}_{a_1,b}\left[a_1^2 \frac{\|Gb\|^2}{r^2}\right] \geq \frac{1}{3 \cdot 12^n} \frac{n\lambda_{\min}^2(G)\mu^2\sigma^2}{(1+4n\lambda_{\max}(G)\sigma^2)^2}.$$

Let $\mu = 1/\sigma^2$ and we have

$$\lim_{\sigma \to 0^+} \mathbb{E}\left[a_1^2 \frac{\|Gb\|^2}{r^2}\right] \geq \lim_{\sigma \to 0^+} \mathbb{E}_{A,x}\mathbb{E}_{a_1,b}\left[\frac{1}{3 \cdot 12^n} \frac{n\lambda_{\min}^2(G)}{\sigma^2(1+4n\lambda_{\max}(G)\sigma^2)^2}\right]$$

$$= \mathbb{E}_{A,x}\mathbb{E}_{a_1,b} \lim_{\sigma \to 0^+}\left[\frac{1}{3 \cdot 12^n} \frac{n\lambda_{\min}^2(G)}{\sigma^2(1+4n\lambda_{\max}(G)\sigma^2)^2}\right]$$

$$= +\infty,$$

where we switch the order of expectation and limit using the monotone convergence theorem. Taking full expectation over $A, x, b$ and $a_1$ of (15) and using the assumption that $\mathbb{E}\|v\|^2 < +\infty$ we have

$$L_{d+1} - L_d = \mathbb{E}_{A,x,b}\|(I-bu)^\top v\|^2 + \mathbb{E}\left[a_1^2 \frac{\|Gb\|^2}{r^2}\right] - \mathbb{E}_{A,x}\|v\|^2 \to +\infty$$

as $\sigma \to 0^+$.

### C.5  Proof of Theorem 10

If we define $G \triangleq (AA^\top)^{-1} \in \mathbb{R}^{n \times n}$ and $u \triangleq \frac{b^\top G}{1+b^\top Gb} \in \mathbb{R}^{1 \times n}$, Lemma 6 implies

$$\begin{bmatrix} A^\top \\ b^\top \end{bmatrix}^+ = \left[(I-bu)^\top(A^+)^\top, u^\top\right].$$

It follows that

$$[A,b]^+[A,b] = \begin{bmatrix} A^+A - \frac{ww^\top}{r} & \frac{w}{r} \\ \frac{w^\top}{r} & 1 - \frac{1}{r} \end{bmatrix},$$

where

$$w = A^+b, \quad r = 1 + b^\top Gb.$$

We obtain the expression for $\mathcal{E}_{d+1}$:

$$\mathcal{E}_{d+1} = \left([x^\top, a_1] \begin{bmatrix} A^\top A - \frac{ww^\top}{r} - I & \frac{w}{r} \\ \frac{w^\top}{r} & -\frac{1}{r} \end{bmatrix} \begin{bmatrix} \beta \\ \beta_1 \end{bmatrix}\right)^2,$$

$$= \left[x^\top\left(A^+A - \frac{ww^\top}{r} - I\right)\beta + \frac{yw^\top\beta}{r} + \frac{x^\top w\beta_1}{r} - \frac{a_1\beta_1}{r}\right]^2$$

$$= \left[x^\top(A^+A - I)\beta + \frac{1}{r}\left(-x^\top ww^\top\beta + x^\top w\beta_1 + a_1 w^\top\beta - a_1\beta_1\right)\right]^2.$$

If $a_1, b_1, \ldots, b_n \overset{iid}{\sim} \mathcal{N}_{\sigma,\mu}^{\mathrm{mix}}$ or $a_1, b_1, \ldots, b_n \overset{iid}{\sim} \mathcal{N}(0, \sigma^2)$, it holds that $\mathbb{E}[a_1] = 0 \in \mathbb{R}$, $\mathbb{E}[x] = 0 \in \mathbb{R}^d$, and $\mathbb{E}[b] = 0 \in \mathbb{R}^{n \times 1}$. Therefore we have

$$\mathbb{E}\left[x^\top \left(A^+ A - I\right) \beta \frac{1}{r} x^\top w \beta_1\right] = \mathbb{E}\left[\frac{1}{r} x^\top \left(A^+ A - I\right) \beta x^\top w\right] \mathbb{E}[\beta_1] = 0,$$

$$\mathbb{E}\left[x^\top \left(A^+ A - I\right) \beta \frac{1}{r} a_1 w^\top \beta\right] = \mathbb{E}\left[x^\top \left(A^+ A - I\right) \beta \frac{1}{r} \mathbb{E}[a_1] w^\top \beta\right] = 0,$$

$$\mathbb{E}\left[x^\top \left(A^+ A - I\right) \beta \frac{1}{r} a_1 \beta_1\right] = \mathbb{E}\left[x^\top \left(A^+ A - I\right) \beta \frac{1}{r} \mathbb{E}[a_1] \beta_1\right] = 0.$$

It follows that

$$\mathbb{E}[\mathcal{E}_{d+1}] = \mathbb{E}\left[x^\top (A^+ A - I)\beta\right]^2 + \mathbb{E}\left[\frac{1}{r^2}\left(-x^\top w w^\top \beta + x^\top w \beta_1 + a_1 w^\top \beta - a_1 \beta_1\right)^2\right]$$

$$+ \mathbb{E}\left[\frac{2}{r} x^\top (A^+ A - I)\beta(-x^\top w w^\top \beta)\right],$$

which then gives

$$\mathbb{E}[\mathcal{E}_{d+1}] - \mathbb{E}[\mathcal{E}_d]$$

$$= \mathbb{E}\left[\frac{1}{r^2}\left(-x^\top w w^\top \beta + x^\top w \beta_1 + a_1 w^\top \beta - a_1 \beta_1\right)^2\right] + \mathbb{E}\left[\frac{2}{r} x^\top (A^+ A - I)\beta(-x^\top w w^\top \beta)\right].$$

First, we consider the second term $\mathbb{E}\left[\frac{2}{r} x^\top (A^+ A - I)\beta(-x^\top w w^\top \beta)\right]$. Note that

$$\mathbb{E}\left[\frac{2}{r} x^\top (A^+ A - I)\beta(-x^\top w w^\top \beta)\right]$$

$$= \mathbb{E}\left[-\frac{2}{r} x^\top (A^+ A - I)\beta\beta^\top w w^\top x\right]$$

$$= \mathbb{E}\left[\frac{2}{r} x^\top (I - A^+ A)\mathbb{E}[\beta\beta^\top] w w^\top x\right]$$

$$= \rho^2 \mathbb{E}\left[\frac{2}{r} x^\top (I - A^+ A) w w^\top x\right],$$

where the second equality is because $\beta$ is independent from the remaining random variables and the third step is because of $\beta \sim \mathcal{N}(0, \rho^2 I)$. Recalling that $w = A^+ b$ and $A^+ A A^+ = A^+$, we have

$$\mathbb{E}\left[\frac{2}{r} x^\top (A^+ A - I)\beta(-x^\top w w^\top \beta)\right]$$

$$= \rho^2 \mathbb{E}\left[\frac{2}{r} x^\top (I - A^+ A) A^+ b w^\top x\right]$$

$$= \rho^2 \mathbb{E}\left[\frac{2}{r} x^\top (A^+ - A^+ A A^+) b w^\top x\right]$$

$$= 0.$$

Now we consider the first term $\mathbb{E}\left[\frac{1}{r^2}\left(-x^\top w w^\top \beta + x^\top w \beta_1 + a_1 w^\top \beta - a_1 \beta_1\right)^2\right]$. Note that all the cross terms vanishes since $\mathbb{E}[\beta] = 0$ and $\mathbb{E}[\beta_1] = 0$. This implies

$$\mathbb{E}\left[\frac{1}{r^2}\left(-x^\top w w^\top \beta + x^\top w \beta_1 + a_1 w^\top \beta - a_1 \beta_1\right)^2\right]$$

$$= \mathbb{E}\left[\frac{1}{r^2}\left((x^\top w w^\top \beta)^2 + (x^\top w \beta_1)^2 + (a_1 w^\top \beta)^2 + (a_1 \beta_1)^2\right)\right]$$

$$= \mathbb{E}\left[\frac{1}{r^2}\left(\mathrm{tr}(xx^\top w w^\top \beta\beta^\top w w^\top) + \beta_1^2(x^\top w w^\top x) + a_1^2 \mathrm{tr}(w w^\top \beta\beta^\top) + a_1^2 \beta_1^2\right)\right]$$

$$= \mathbb{E}\left[\frac{1}{r^2}\left(\rho^2 \|w\|^2 \mathrm{tr}(xx^\top w w^\top) + \rho^2(x^\top w w^\top x) + a_1^2 \rho^2 \|w\|^2 + a_1^2 \rho^2\right)\right]$$

$$= \rho^2 \mathbb{E}\left[\frac{1}{r^2}(\|w\|^2 + 1)((x^\top w)^2 + \mathbb{E}[a_1^2])\right],$$

where the third equality is because of $[\beta^\top, \beta_1]^\top \sim \mathcal{N}(0, \rho^2 I_{d+1})$. From the above calculation one can see that $\mathbb{E}[\mathcal{E}_{d+1}] > \mathbb{E}[\mathcal{E}_d]$.

If $a_1, b_1, \ldots, b_n \overset{iid}{\sim} \mathcal{N}_{\sigma,\mu}^{\mathrm{mix}}$, Theorem 9 implies that for any $C > 0$, there exist $\mu, \sigma$ such that

$$
\mathbb{E}\left\| \begin{bmatrix} A^\top \\ b^\top \end{bmatrix}^+ \begin{bmatrix} x \\ a_1 \end{bmatrix} \right\|^2 - \mathbb{E}\left\| (A^+)^\top x \right\|^2 > C \, .
$$

Because $\mathbb{E}[\mathcal{E}_{d+1}] \geq \mathbb{E}[\mathcal{E}_d]$, we obtain that for any $C > 0$, there exist $\mu, \sigma$ such that $L_{d+1}^{\mathrm{exp}} - L_d^{\mathrm{exp}} > C$.

If $a_1, b_1, \ldots, b_n \overset{iid}{\sim} \mathcal{N}(0, \sigma^2)$, we have as $\sigma \to 0$,

$$
\mathbb{E}[\mathcal{E}_{d+1}] - \mathbb{E}[\mathcal{E}_d] = \rho^2 \sigma^2 \mathbb{E}\left[ \frac{1}{r^2}(\sigma^2 \|A^+\|^2 + 1)(\|A^{+\top} x\|^2 + 1) \right] \sim \rho^2 \sigma^2 \left( \mathbb{E}\|A^{+\top} x\|^2 + 1 \right) \, .
$$

From the proof of Theorem 8, we know that as $\sigma \to 0^+$

$$
\left\| \begin{bmatrix} A^\top \\ b^\top \end{bmatrix}^+ \begin{bmatrix} x \\ a_1 \end{bmatrix} \right\|^2 - \mathbb{E}\left\| (A^\top)^+ x \right\|^2 \sim -2\sigma^2 \mathbb{E}[\|(A^\top A)^+ x\|^2] \, .
$$

If $\rho \leq \eta \sqrt{\frac{\mathbb{E}[\|(A^\top A)^+ x\|^2]}{\mathbb{E}\|A^{+\top} x\|^2 + 1}}$, we have

$$
L_{d+1}^{\mathrm{exp}} - L_d^{\mathrm{exp}} \sim -\sigma^2 \left( 2\eta^2 \mathbb{E}[\|(A^\top A)^+ x\|^2] - \rho^2 \left( \mathbb{E}\|A^{+\top} x\|^2 + 1 \right) \right) \leq -\sigma^2 \eta^2 \mathbb{E}[\|(A^\top A)^+ x\|^2] \, .
$$

As a result, there exists $\sigma > 0$ such that for all $\rho \leq \eta \sqrt{\frac{\mathbb{E}[\|(A^\top A)^+ x\|^2]}{\mathbb{E}\|A^{+\top} x\|^2 + 1}}$, we have $L_{d+1}^{\mathrm{exp}} < L_d^{\mathrm{exp}}$.

## D  Discussion

Recently, there has been growing interest in the comparison and connection between deep learning and classical machine learning methods. For example, clustering, a classical unsupervised machine learning method, was adapted to end-to-end training of image data [17, 24–27]. This paper studied the non-monotonic generalization risk curve of overparametrized linear regression. It would be an interesting future work to study the multiple descent phenomenon in other classical machine learning methods and theoretically understand this phenomenon in deep learning. Moreover, when the multiple descent phenomenon arises in different machine learning models, it remains open whether there is any deep reason in common that accounts for it.