# OpenReview forum: "Multiple Descent: Design Your Own Generalization Curve"
_NeurIPS.cc/2021/Conference — NeurIPS 2021 Poster_

### Official Review · Reviewer_qLTq · 2021-07-09

**Rating:** 6
**Confidence:** 4

**Summary:**

I have read other reviewers' comments and the author response.
I lower my score due to the following reasons: 1) changing the feature dimension $d$ doesn't change the groundtruth function, which makes the problem setting questionable. 2) the response to the issue "\sigma should be taken far smaller than O(1/5^n)" is unsatisfactory, which makes the result unattainable.
=============================================
This paper studies the multiple descent of the excess risk for linear regression in under/over-parameterized regimes. In the under-parameterized regime, Theorem 3.1 includes two parts: one is showing that the loss $L_d$ non-decreases with $d$ independent of the data distribution; the other is $L_{d+1} - L_d > C$. In the over-parameterized regime, the authors provide the multiple descents in Theorem 5 under $\beta = 0$ and Theorem 11 with $\beta$ being Gaussian.

**Limitations And Societal Impact:**

yes

**Main Review:**

The quality of this paper is good but there are several points that I concern:

**Problem settings:**

The target function is assumed to be linear, low-dimensional ($d<D$), which is fair and standard in interpolation learning.
However, it appears a little strange to me when the authors consider a linear regression over its partial features (i.e., $d<D$) so as to tune $d$. I do understand that, in linear regression, adding one dimension to features is equivalent to increasing the model complexity (strictly speaking, at least non-decreasing). This operation/setting can be also found in [51]. However, a learning problem often corresponds to a given/fixed d. A common way is to tune the number of samples n instead of d, which is more reasonable in practice. So is it possible to transform the results via tuning $d$ in this paper to the standard tuning $n$ setting, e.g., [Liu, Liao, and Suykens AISTATS2021] on kernel regression in high dimensions?


**Technical details:**

The proof in this paper largely depends on taking $\sigma  \rightarrow 0_+$. This might be unattainable in some cases. Here I take one example to illustrate this.
I’m questionable the second part in Theorem 3.1, i.e., the proof of Theorem 4.
First, the constant $C$ in fact depends on $\sigma$?
Second, the employed strategy in the proof is to derive the lower bound in line 600 such that the second term in line 595 is sufficiently large. To this end, the authors derive this lower bound as the exponential decreasing order of n. In this case, \sigma should be taken far smaller than $O(1/5^{n})$ if we want to ensure the non-negativity of line 599. However, this might be unattainable in practice. For example, if we only consider n=100 samples, sigma should be taken as 1e-70 order. Accordingly, this condition should be better re-organized. In theory, it could be possible to derive a polynomial order (at most) of $n$ in line 599? In experiment, Theorem 4 can be experimentally validated on some synthetic datasets: obtaining A and x, and seeking for potential solutions of $C$ and $\sigma$.


**Minor issues:**

1．	The authors claim that “To our best knowledge, this is the first work proving a multiple descent phenomenon for any learning model.” In line 67. I suggest the authors polish this sentence if [34] is taken into consideration.


**Time Spent Reviewing:**

4

---

> ### Author Response · Authors · 2021-08-10
> **Response to Reviewer qLTq**
>
> Thank you for your positive comments!
>
> ---
> **Q1**: The target function is assumed to be linear, low-dimensional ($d<D$), which is fair and standard in interpolation learning. However, it appears a little strange to me when the authors consider a linear regression over its partial features (i.e., $d<D$) so as to tune $d$. I do understand that, in linear regression, adding one dimension to features is equivalent to increasing the model complexity (strictly speaking, at least non-decreasing). This operation/setting can be also found in [51].
>
> **A1**: As mentioned by the reviewer, this setting can be also found in [51]. This setup was also used in previous influential works in this field, e.g., Equations (1) and (2) in Hastie et al. [28] and Liang et al. [34] (if the kernel is set to the linear kernel). We discussed it in line 114-116 and will further clarify it in the revision.
>
> ---
> **Q2**: Is it possible to transform the results via tuning $d$ in this paper to the standard tuning $n$ setting, e.g., [Liu, Liao, and Suykens AISTATS2021] on kernel regression in high dimensions?
>
> **A2**: We thank the reviewer for the great point and for the reference [Liu, Liao, and Suykens AISTATS2021]. This is a very relevant paper and we will cite and discuss it in the revision. In fact, $n$ and $d$ are not equivalent. Increasing $n$ means adding a new IID sample, while increasing $d$ means adding a new entry to each IID sample. [38] and [39] studied tuning $n$. [38] is a mostly empirical paper. [39] is a theory paper and their techniques for analyzing tuning $n$ are different from the works that study tuning $d$.
>
> ---
> **Q3**: The proof in this paper largely depends on taking $\sigma \to 0_+$. This might be unattainable in some cases.
> Detail: Here I take one example to illustrate this. I’m questioning the second part in Theorem 3.1, i.e., the proof of Theorem 4. First, the constant C in fact depends on $\sigma$? Second, the employed strategy in the proof is to derive the lower bound in line 600 such that the second term in line 595 is sufficiently large. To this end, the authors derive this lower bound as the exponential decreasing order of n. In this case, $\sigma$ should be taken far smaller than $O(1/5^n)$ if we want to ensure the non-negativity of line 599. However, this might be unattainable in practice.
>
> **A3**: The reviewer’s idea on how our proof works is totally correct. We agree with the reviewer that $\sigma\to 0$ does not happen in most real data. However, we emphasize that to the best of our knowledge, this paper shows the first provable existence result of multiple descent and that indeed the shape of the generalization curve can be designed to some degree, so that researchers and practitioners  will have a more comprehensive view of what the generalization curve could look like.
>
> ---
> **Q4**: The authors claim that “To our best knowledge, this is the first work proving a multiple descent phenomenon for any learning model.” In line 67. I suggest the authors polish this sentence if [34] is taken into consideration.
>
> **A4**: [34] presented an upper bound on the risk of the minimum-norm interpolation versus the data dimension in reproducing kernel Hilbert spaces (RKHS). The upper bound exhibits multiple descents. However, a multiple-descent upper bound without a properly matching lower bound does NOT imply the existence of a multiple-descent generalization curve, and the authors clearly stated this fact in their own paper. We discussed this in Line 53 of our paper. However, [34] did show empirically the multiple descent phenomenon. As suggested by the reviewer, in the revision, we will emphasize that we prove multiple descent theoretically and [34] showed it empirically.

---

> > ### Author Response · Authors · 2021-08-24
> > **Response to Reviewer qLTq's follow-up**
> >
> > We thank the reviewer for the follow-up. We believe there is some misunderstanding and we would like to clarify here.
> >
> > (1) This is a common setup in the literature. In lines 114-116, we clarified that the same setup was used in prior influential works. For example, see Equations (1) and (2) in Hastie et al. [28] and Liang et al. [34] (if the kernel is set to the linear kernel).
> >
> > (2) Our construction is not meant to model a specific practical setting but rather a demonstration how multiple descents can arise. In line 84, we have mentioned that “we rarely observe complex generalization curves in practice”. Our construction provided a possible explanation for its rare occurrence.
> >
> > We hope our response addresses the follow-up questions of the reviewer.

---

> > > ### Author Response · Authors · 2021-08-27
> > > **Response to the question regarding our setup**
> > >
> > > For our setup, we would like to emphasize again that our setup is the same as [34] (if the kernel is set to the linear kernel) and that it is indeed a common setup in the literature. The $d$ in [34] is **NOT** fixed. They wrote on Page 5 "Our investigation focuses on the regime when the dimension $d(n)$ grows with the sample size $n$, with $n$ being sufficiently large." (the sentence right after Equation (5) on Page 5). For another example, see Figure 2 on page 3: The sample size $n=5000$ is fixed and the x-axis is $\log d$.
> > >
> > > We are happy to answer further questions on this point. Thank you!

---

### Official Review · Reviewer_sF9e · 2021-07-12

**Rating:** 3
**Confidence:** 4

**Summary:**

This paper studies the generalization error of the min-l2 norm solutions of linear regression. Authors claim that their analytical results can explain the multiple peaks in the generalization curve with respect to the number of features. However, the system model and some fundamental steps of the analysis are problematic, which makes me doubt the correctness of the paper.

**Limitations And Societal Impact:**

Yes.

**Main Review:**

The problem setup in this paper seems not correct. Generally speaking, the output of the training data y is given beforehand and shouldn’t depend on the number of features d (or any other manually chosen parameters) used in the linear regression. However, the problem setup on page 3 states that y=\tilde{x}^T\beta+noise, where \tilde{x}=x[1:d] depends on d, and thus y depends on d. That makes the generalization error curve with respect to d in this paper useless because the error curve is not for one specific ground-truth function but for different ground-truth functions when d changes. In order to study the effect of the number of features on the generalization error, a more correct setup should be y=x^T\beta+noise instead of y=\tilde{x}^T\beta+noise. Authors can take the following paper as an example: “M. Belkin, D. Hsu, and J. Xu. Two models of double descent for weak features. arXiv preprint arXiv:1903.07571, 2019.”

In Eq.(1) (the generalization error), it is unclear what random variable (e.g., training input, test input, noise in the training input) in each step the expectation is based on. Besides, the second step of Eq.(1) seems incorrect, i.e. E[(y-x^T\hat{beta})^2-(y-x^T\beta)^2] is not equal to E[(x^T(\hat{beta}-\beta))^2].

Remark 1 is incorrect. Authors claim that A^+A=I, which is wrong. I think the correct one should be E[A^+A]=I where the expectation is on the design matrix A (i.e., the training input x).

Remark 1 implies that the bias term is always zero in the under-parameterized region and authors completely discard the bias term in Eq.(2). This contradicts the well-known bias-variance tradeoff curve (the U-shape curve) in the under-parameterized region of linear regression. Similarly, Theorem 1 says that L_d is always increasing with respect to d, which also contradicts the known U-shape curve. The authors explain on page 5 that a "U-shaped curve does not always occur" and use the Prop.2 of [28] as evidence. However, Prop.2 of [28] describes the situation that the number of features and the number of training input go to infinity at a constant ratio, which is not relevant to the situation in this paper where both the number of features and the number of training inputs are finite. Besides, I haven't seen any literature (including [28]) that says "U-shape curve does not occur in the under-parameterized region". Indeed, "U-shape curve does not occur" is a very strong statement and should deserve more careful justification. For example, authors should clearly point out why the classical derivation of bias-variance tradeoff does not hold here. Is it because this paper uses a different setup? Or is it because the classical derivation of the bias-variance tradeoff is totally wrong?

Overall, the setup and some fundamental steps of the derivation seems problematic and the main result contradicts the well-known bias-variance tradeoff result in the under-parameterized region, which jeopardizes the credibility and the contribution of this paper.

**Time Spent Reviewing:**

4

---

> ### Author Response · Authors · 2021-08-10
> **Response to Reviewer sF9e**
>
> Thank you for the feedback. It looks that there are a few misunderstandings in your review. We will explain it in detail in the following.
>
> ---
> **Q1**: The problem setup in this paper seems not correct. Generally speaking, the output of the training data y is given beforehand and shouldn’t depend on the number of features d (or any other manually chosen parameters) used in the linear regression.
> Authors can take the following paper as an example of the correct model: “M. Belkin, D. Hsu, and J. Xu. Two models of double descent for weak features. arXiv preprint arXiv:1903.07571, 2019.”
>
> **A1**: This is a common setup in the literature. In lines 114-116, we clarified that the same setup was used in prior influential works. For example, see Equations (1) and (2) in Hastie et al. [28] and Liang et al. [34] (if the kernel is set to the linear kernel).
>
> ---
> **Q2**: In Eq.(1) (the generalization error), it is unclear what random variable (e.g., training input, test input, noise in the training input) in each step the expectation is based on.
>
> **A2**: The expectation is taken over the randomness of training input, test input and noise. This is standard, see, for example, Theorem 1 and Corollary 1 of Belkin et al. [11]. Belkin et al. [11] wrote explicitly in Corollary 1, “the risk of $\hat{\beta}$ (taking expectation with respect to ... the random design matrix and response vector)”. The random design matrix $X$ and response vector $y$ are the training input, which includes the randomness in the noise. Their expectation is also taken over the test input and noise therein.
>
> ---
> **Q3**: Besides, the second step of Eq.(1) seems incorrect, i.e. $E[(y-x^T\hat{\beta})^2-(y-x^T\beta)^2]$ is not equal to $E[(x^T(\hat{\beta}-\beta))^2]$.
>
> **A3**: Our equation is correct and a basic result in linear regression. The same equation also appears in the long version of Bartlett et al. [8], see the first equation in the proof of Lemma 18 on page 20.
>
> Long version of Bartlett et al. [8]: Bartlett, P. L., Long, P. M., Lugosi, G., & Tsigler, A. (2019). Benign Overfitting in Linear Regression. arXiv preprint arXiv:1906.11300.
>
> ---
> **Q4**: Remark 1 is incorrect. Authors claim that $A^+ A=I$, which is wrong. I think the correct one should be $\mathbb{E}[A^+A]=I$ where the expectation is on the design matrix $A$ (i.e., the training input $x$).
>
> **A4**: Our equation $A^+ A=I$ is indeed correct. When $A$ has linearly independent columns as in our paper, its pseudoinverse is a left inverse, i.e., $A^+ A=I$. We don’t need the expectation.
>
> ---
> **Q5**: Remark 1 implies that the bias term is always zero in the under-parameterized region and authors completely discard the bias term in Eq.(2). This contradicts the well-known bias-variance tradeoff curve (the U-shape curve) in the under-parameterized region of linear regression. Similarly, Theorem 1 says that $L_d$ is always increasing with respect to $d$, which also contradicts the known U-shape curve. The authors explain on page 5 that a "U-shaped curve does not always occur" and use the Prop.2 of [28] as evidence. However, Prop.2 of [28] describes the situation that the number of features and the number of training input go to infinity at a constant ratio, which is not relevant to the situation in this paper where both the number of features and the number of training inputs are finite.
>
> **A5**: First, the result that the bias term is always zero does NOT rely on whether $n$, $d$ are finite or going to infinity. It follows from the basic result that $A^+ A = I$ if $A$ has linearly independent columns (this is the case almost surely in the underparameterized case if $A$ has a continuous distribution).
> Second, in our paper, we have already explained that Theorem 1 conveys the same picture as Prop. 2 of [28]. Although in [28] the number of features and the training input go to infinity at a constant ratio $\gamma$, the ratio $\gamma$ reflects how large $d$ is compared to $n$. Prop. 2 of [28] shows that there is no bias (see the sentence right after Prop. 2 of [28]: “​​As it can be seen from the last proposition, in the underparametrized case the risk is just variance”). See the three “min-norm LS” curves in Fig. 1of [28], all three curves are increasing when $d/n \to \gamma < 1$ (i.e., the underparameterized regime). While [28] is an asymptotic result, it holds when we pick a very large $n$, and change $d$ from 1 to a very large value, and then $\gamma = d/n$ varies from 0 to $\infty$.
> Again, we emphasize that the result that the bias term is zero does NOT rely on whether $n, d$ are finite or going to infinity.
>
> ---
> **Q6**: Besides, I haven't seen any literature (including [28]) that says "U-shape curve does not occur in the under-parameterized region". Indeed, "U-shape curve does not occur" is a very strong statement and should deserve more careful justification. For example, authors should clearly point out why the classical derivation of bias-variance tradeoff does not hold here. Is it because this paper uses a different setup? Or is it because the classical derivation of the bias-variance tradeoff is totally wrong?
>
> **A6**: What we wrote is “the U-shaped curve does not ALWAYS occur.” (line 173). The goal of this paper is to show that there are exceptions to the classical U-shaped curve, and this is what other double descent papers try to show, and the double descent curve itself is, of course, not a U-shaped curve. We have provided concrete construction and proofs in our paper to point out why the classical U-shaped curve does not hold here.
>
> ---
> We hope the above response addresses your concerns.

---

> ### Author Response · Authors · 2021-08-31
> **Would you please check our response?**
>
> Dear Reviewer sF9e,
>
> It would be great if you would check our response to see if we addressed your concerns. We believe there are some misunderstandings and have clarified them in the response. We would be glad to answer any further questions that you have.
>
> Many thanks for your time!
>
> Best
> Authors

---

### Official Review · Reviewer_wnTz · 2021-07-16

**Rating:** 7
**Confidence:** 3

**Summary:**

The authors systematically study a simple linear regression setup where an arbitrary number of ascents and descents can be obtained in the risk.

**Limitations And Societal Impact:**

Theory work without a clear societal impact.

**Main Review:**

Originality: Multiple-descent curves are studied in several works as cited in the paper. As far as I know, this one is unique in proposing an algorithmic construction of input data distribution.

Significance: It is an interesting example setup that anyone who works on this subject should keep in mind.

Clarity: The paper is written very clearly and has a good structure.

Quality: Although I could not check the appendix due to time constraints, I did not detect any issues in theorems neither proofs.

I have two comments:
- line 68: "for any learning model". The authors consider a very constrained setup of linear regression, not any general learning model.
- line 311: Conjecture for neural networks. In my opinion, a conjecture needs some level of motivation. This sentence seems very off. In the paper, the model size grows as the input data dimensionality grows. In neural networks, the model size is totally independent of the data. The more obvious difference is non-linearity. Please remove this conjecture, or justify it with regards to your setup.

**Time Spent Reviewing:**

2

---

> ### Author Response · Authors · 2021-08-10
> **Response to Reviewer wnTz**
>
> Thank you for your positive comments!
>
> ---
> **Q1**: line 68: "for any learning model". The authors consider a very constrained setup of linear regression, not any general learning model.
>
> **A1**:  Thank you for pointing it out. We will remove "for any learning model" in the revision.
>
> ---
>
> **Q2**: line 311: Conjecture for neural networks. In my opinion, a conjecture needs some level of motivation. This sentence seems very off. In the paper, the model size grows as the input data dimensionality grows. In neural networks, the model size is totally independent of the data. The more obvious difference is non-linearity. Please remove this conjecture, or justify it with regards to your setup.
>
> **A2**: Thank you for the great point! We will remove this sentence from the conclusion section.

---

> > ### Comment · Reviewer_wnTz · 2021-08-27
> > **Setup at reference [34]**
> >
> > Dear authors,
> >
> > Thanks for your reply. I read the other reviewers' comments. It seems that the reference [34] does not use the same setup: i.e. they have a fixed true function $f^*$ and fixed $ d $ (so the data generating process is fixed), and they study the behavior as the number of samples $ n = d^{1/\alpha} $ change. Can you please comment on this point?

---

> > > ### Author Response · Authors · 2021-08-27
> > > **Response to Reviewer wnTz's follow-up question**
> > >
> > > The $d$ in [34] is **NOT** fixed. They wrote on Page 5 "Our investigation focuses on the regime when the dimension $d(n)$ grows with the sample size $n$, with $n$ being sufficiently large." (the sentence right after Equation (5) on Page 5). For another example, see Figure 2 on page 3: The sample size $n=5000$ is fixed and the x-axis is $\log d$. We would like to emphasize that our setup is the same as [34] (if the kernel is set to the linear kernel) and that it is indeed a common setup in the literature.
> > >
> > > We are happy to answer further questions on this point. Thank you!

---

### Official Review · Reviewer_w6vd · 2021-07-21

**Rating:** 7
**Confidence:** 3

**Summary:**

=================================
FINAL UPDATE AFTER DISCUSSION PHASE

As the authors addressed the points that came up during the discussion phase (which made me initially lower the score), I revert back to my initial scoring of accept.


===================================
UPDATE AFTER INITIAL DISCUSSION PHASE

The authors addressed my own points, but following the discussion with the other reviewers I decided to lower my score. It was pointed out that the data generation process depends on the model-dimension, and thus the resulting investigation is not about 'generalization curves' as claimed by the authors, but rather about the generalization behavior of different interpolation regimes. While I believe the contribution is still valuable, some explanations and need be changed (which may be possible in a camera ready version).

( For example I don't believe anymore that the paper indeed answers the question: 'Can the existence of a multiple descent generalization curve be rigorously proven?'.)

==================================

The paper analyses the behavior of generalization curves of linear regression wrt an increasing feature dimensionality. The paper constructs specific examples that can generate any ascending/ descending behavior of the curve, for both the under and overparametrized case. Their construction highlights that ascending/ descending behavior of the generalization curve is not inherent to the model family, but rather a result of the data distribution in combination with the model.

**Limitations And Societal Impact:**

Yes.

**Main Review:**

Summary review:

Strength:
+ Fairly polished
+ Main story is easy enough to follow, despite some parts being rather technical
+ Original
+ Technically sound (checked some proofs of Section 3, not from 4)

Weaknesses:
- Some remarks are missed on me
- No discussion

Detailed review:

The paper considers learning curves, a topic that recently gained traction again. Previous works report various types of multiple descending behaviors, either empirical or theoretical, but this works provides, to the best of my knowledge, the most rigorous and general example that highlights this behavior. As I don't see any major flaws, I will thus vote to accept the paper.

There are some remarks in the paper that got missed on me:

Remark 2: In particular the part 'For the second part ... we can change the order of the features'.
At this stage of the paper it is pretty unclear how the product relates to C. But more importantly: We can change the order of the features, but what is the consequence of that/ why is that an interesting observation?

Remark 3: I missed a bit the point of the remark. Is it just to demonstrate that sampling from $N^{mix}_{\sigma,1}$ is not difficult, or is there more to it?

Line 181: $x_1$ is not the best choice for the additional feature of the test vector as $x_1$ is also a training sample. For me sticking to the $b$ notation, e.g. $b^0$ or $b^{test}$, would be better to follow.

The Gaussian $\beta$ setting:

It is a bit curious to me that we have now a distribution of $\beta$. I did not check the proof of Theorem 10, but from the statements it seems like it also works for a single $\beta$, by looking at the statements for $\rho \to 0$. It would be good if you could comment on that. Do we need it? Is it there to be more general? What do we lose/maintain if we assume a single $\beta$.

I personally find the conclusion a bit weak. I would like to see more of a discussion. You conjecture that multiple descents also occur for NN, is that based on any observation or intuition? In the beginning you state that those curves are rarely seen in practice. Does your work now shed a new light on that observation? Or at least point into a new direction to tackle it?




**Time Spent Reviewing:**

4

---

> ### Author Response · Authors · 2021-08-10
> **Response to Reviewer w6vd**
>
> Thank you for your positive comments!
>
> ---
> **Q1**: Remark 2: In particular the part 'For the second part ... we can change the order of the features'. At this stage of the paper it is pretty unclear how the product relates to $C$. But more importantly: We can change the order of the features, but what is the consequence of that/why is that an interesting observation?
>
> **A1**: Thanks for pointing this out. We wrote this sentence to highlight that our construction is a product distribution. Therefore, the marginal contribution of each entry is independent, does not depend on each other, and therefore is interchangeable. If some entries follow a conditional distribution on others, then they are not interchangeable. This remark that we made in the paper may be minor. We will revise our wording or remove this remark to make the paper clearer.
>
> ---
> **Q2**: Remark 3: I missed a bit the point of the remark. Is it just to demonstrate that sampling from $N^{mix}_{\sigma,1}$ is not difficult, or is there more to it?
>
> **A2**: Thank you for the great point! The point of Remark 3 is that while our construction needs the data distribution to be Gaussian or Gaussian mixture, we can also simulate it by using Gaussian data only and a non-linear kernel (the kernel will distort the Gaussian data and simulate Gaussian mixture). We will elaborate on it in the final revision.
>
> ---
> **Q3**: Line 181: $x_1$ is not the best choice for the additional feature of the test vector as $x_1$ is also a training sample. For me sticking to the $b$ notation, e.g. $b^0$ or $b^{test}$, would be better to follow.
>
> **A3**: Thank you for the suggestion! We will make the notation clearer.
>
> ---
>
> **Q4**: It is a bit curious to me that we now have a distribution of $\beta$. I did not check the proof of Theorem 10, but from the statements it seems like it also works for a single $\beta$, by looking at the statements for $\rho \to 0$. It would be good if you could comment on that. Do we need it? Is it there to be more general? What do we lose/maintain if we assume a single $\beta$.
>
> **A4**: The reviewer touches on a very important part of the proof. We considered Gaussian $\beta$ because we would like to study the case where $\beta$ is non-zero. To this end, we need to choose a specific $\beta$ and specify its value in each entry, which could be restrictive. By considering Gaussian $\beta$, we studied the curve of the Bayesian risk with a Gaussian prior. We believe that this setup could be more practically interesting. We did not see a straightforward way of transferring the result for the Bayesian risk to the result for an individual $\beta$. Thus we need the Bayesian risk here. We will elaborate on it in the revision.
>
> ---
>
> **Q5**: I personally find the conclusion a bit weak. I would like to see more of a discussion. You conjecture that multiple descents also occur for NN, is that based on any observation or intuition? In the beginning you state that those curves are rarely seen in practice. Does your work now shed a new light on that observation? Or at least point into a new direction to tackle it?
>
> **A5**: Thank you for the great point! We will remove this sentence from the conclusion section.  We believe that showing multiple descent in linear regression is a good starting point for explaining non-monotonic generalization curves observed in machine learning studies. In the revision, we will further discuss whether multiple descent will happen in more general learning models.

---

> > ### Author Response · Authors · 2021-08-27
> > **Response to Reviewer w6vd's updated review**
> >
> > For our setup, we would like to emphasize again that our setup is the same as [34] (if the kernel is set to the linear kernel) and that it is indeed a common setup in the literature. The $d$ in [34] is **NOT** fixed. They wrote on Page 5 "Our investigation focuses on the regime when the dimension $d(n)$ grows with the sample size $n$, with $n$ being sufficiently large." (the sentence right after Equation (5) on Page 5). For another example, see Figure 2 on page 3: The sample size $n=5000$ is fixed and the x-axis is $\log d$.
> >
> > We are happy to answer further questions on this point. Thank you!

---

> > > ### Comment · Reviewer_w6vd · 2021-08-30
> > > **Response**
> > >
> > > Yes I do agree with the setup being the same, but from what I can see [34] does not claim to analyze 'generalization curves'. That said, they do actually have a remark on that stating "Depending on the point of view, we can also interpret the upper bound of Theorem 1 by fixing d and analyzing the behavior in n. In this case, the interpretation is rather counterintuitive: more data can lead to alternating regimes of better and worse performance."
> > > Although their Theorem 1 is of course only an upper bound, it does predict peaking in an actual generalization curve when d is fixed. I do not see that the same claims can be made with your results, but please correct me in case I am wrong with that or missing something!

---

> > > > ### Author Response · Authors · 2021-09-01
> > > > **Further Clarification for Reviewer w6vd**
> > > >
> > > > We are happy to know that you agree that we share the same setup with [34]. Regarding your further questions:
> > > >
> > > > ---
> > > >
> > > > **Q**: from what I can see [34] does not claim to analyze 'generalization curves'. That said, they do actually have a remark on that stating "Depending on the point of view, we can also interpret the upper bound of Theorem 1 by fixing d and analyzing the behavior in n. In this case, the interpretation is rather counterintuitive: more data can lead to alternating regimes of better and worse performance."
> > > >
> > > > **A**: As they stated (“we can **also** interpret”), this is just an **alternative** point of view. The paragraph right before this alternative interpretation discussed the behavior of the generalization curve in Figure 1 and Theorem 1. Their **primary** goal is indeed to analyze the generalization curve and their Figures 1 and 2 did illustrate the generalization curves.
> > > >
> > > > ---
> > > >
> > > > **Q**: Although their Theorem 1 is of course only an upper bound, it does predict peaking in an actual generalization curve when d is fixed. I do not see that the same claims can be made with your results
> > > >
> > > > **A**: First, we would like to clarify the understanding of Theorem 1 in [34]. In order for a peak to exist, the dimension $d$ can **NOT** be fixed, since the x-axis of the curve is $d$.
> > > >
> > > > Second, we emphasize again that [34] does not have a matching lower bound for the generalization, and that an upper bound without a lower bound **cannot** prove multiple descents for the generalization curve. They admitted this in their paper (page 3 in [34]: “The challenging problem of proving a lower bound that exhibits the multiple descent behavior remains open.”). We have also mentioned this clearly in our paper (see Lines 55-57 in our submission). In our paper, we directly studied the generalization itself (instead of just a bound, which is only a surrogate) and rigorously proved the existence of multiple descents. Our work followed up the unexplored challenges of [34], shared the same setup as theirs, and proved the multiple descents of the generalization curve itself, rather than any surrogate (say, an upper or lower bound).
> > > >
> > > > ---
> > > >
> > > > We hope that you would reconsider your score if our response addresses your questions. Please let us know if you have any further questions.
> > > >
> > > > Best,
> > > >
> > > > Authors

---

### Author Response · Authors · 2021-09-02
**Further response from the authors**

Our setup is indeed the same as that of recent well-cited works, for example [34] and [28]. During the rolling discussions with Reviewer w6vd, Reviewer w6vd has agreed that our setup is the same as [34]. We hope this clarifies the issue. Our work followed up the unexplored challenges of [34], shared the same setup as theirs, and proved the multiple descents of the generalization curve itself, rather than any surrogate (say, an upper or lower bound).

We hope that the explanations and details that we provided in all the responses could address the reviewers' questions. It would be great if the reviewers could take all our responses into consideration.

Thank you for your time and efforts!

Best,

Authors of paper 2228

---

### Decision · Program_Chairs · 2021-09-27

**Decision:**

Accept (Poster)

**Comment:**

We thank the authors for this submission. Overall, the paper presents an interesting perspective on the fact that the generalization curve can have multiple descents, and that the locations can be explicitly controlled.

The paper well-motivates the approach. The authors have provided extensive responses to the concerns raised and the AC + reviewers really thank them for their effort. Overall, the new results obtained during the rebuttal definitely improve the quality of the paper. We all believe that the inclusion of these results during the rebuttal period is something that does not heavily change the message of this paper.

There was discussion and consensus that this work is interesting. However, there are definitely points that need to be handled, based on the discussion so far. Having in mind issues/concerns raised by the reviewers, the main points of reviewers during further discussion were that this paper deserves publication, given the promised fixes by the authors during the discussion period.